# Elevated rates of autism, other neurodevelopmental and psychiatric diagnoses, and autistic traits in transgender and gender-diverse individuals

Varun Warrier [1✉], David M. Greenberg[1,2], Elizabeth Weir [1], Clara Buckingham[1], Paula Smith[1], Meng-Chuan Lai [1,3,4], Carrie Allison[1] & Simon Baron-Cohen[1✉]

It is unclear whether transgender and gender-diverse individuals have elevated rates of autism diagnosis or traits related to autism compared to cisgender individuals in large non-clinic-based cohorts. To investigate this, we use five independently recruited cross-sectional datasets consisting of 641,860 individuals who completed information on gender, neurodevelopmental and psychiatric diagnoses including autism, and measures of traits related to autism (self-report measures of autistic traits, empathy, systemizing, and sensory sensitivity). Compared to cisgender individuals, transgender and gender-diverse individuals have, on average, higher rates of autism, other neurodevelopmental and psychiatric diagnoses. For both autistic and non-autistic individuals, transgender and gender-diverse individuals score, on average, higher on self-report measures of autistic traits, systemizing, and sensory sensitivity, and, on average, lower on self-report measures of empathy. The results may have clinical implications for improving access to mental health care and tailoring adequate support for transgender and gender-diverse individuals.

[1] Autism Research Centre, Department of Psychiatry, University of Cambridge, Douglas House, 18B Trumpington Road, Cambridge CB2 8AH, UK.
[2] Interdisciplinary Department of Social Sciences and Department of Music, Bar-Ilan University, Ramat Gan 5290002, Israel. [3] Child and Youth Mental Health Collaborative, Centre for Addiction and Mental Health and The Hospital for Sick Children, Department of Psychiatry, University of Toronto, 80 Workman Way, Toronto, ON M6J 1H4, Canada. [4] Department of Psychiatry, National Taiwan University Hospital and College of Medicine, No. 7, Zhongshan South Rd., Taipei 10002, Taiwan. ✉email: vw260@medschl.cam.ac.uk; sb205@cam.ac.uk

Autism is a group of neurodevelopmental conditions characterized by early-emerging difficulties in social-communication, unusually repetitive behavior and narrow interests, and atypical sensory sensitivity[1]. Approximately 1–2% of the general population is estimated to be autistic based on large-scale prevalence and surveillance studies, although these numbers vary between countries, age at the time of assessment and other criteria[2–8]. Whilst several studies have investigated rates of autism in individuals who are birth-assigned as males and females, there still is limited information on rates of autism in transgender and gender-diverse individuals in the general population. Gender identity is a different construct from sex assigned at birth, which is typically classified as male or female primarily based on external genitalia. Some individuals are born with chromosomal, genital, or hormonal sex-characteristics which vary from the male–female binary (intersex individuals) and who may be assigned as or raised as males or females. Gender identity is a person's sense of their own gender, which may or may not coincide with sex assigned at birth. Following current recommended practice, we use the term "cisgender" to refer to individuals whose gender corresponds to their sex assigned at birth. However, there is a diversity of gender identities including transgender, non-binary, genderfluid, agender, genderqueer, two-spirit, bigender or others. Again, based on current recommended practice, we collectively refer to these and other diverse gender identities as "transgender and gender-diverse" (i.e., individuals whose gender does not always correspond to the sex they were assigned at birth). Currently, 0.4–1.3%[9–11] of the general population is estimated to be transgender and gender-diverse, although the numbers vary considerably based on how the terms are defined[11].

A few studies, mostly clinic-based, typically with small sample sizes, and in individuals with gender dysphoria (GD, defined as persistent distress arising from a mismatch between sex assigned at birth and gender identity), have investigated the link between autism/traits related to autism and gender diversity[12,13]. These studies have identified increased rates of gender diversity in autistic children and adolescents[14–18], and adults[19,20], compared to the general population. Most of these studies in children and adolescents have used a single item on the Child Behavior Checklist (CBCL), a caregiver-report measure for behavioral problems, to quantify gender variance, and these have identified that between 4% and 5.4% of autistic children may potentially be transgender or gender-diverse, compared to 0.7% of non-autistic children[14–16]. The largest of these, conducted in nearly 300,000 children, identified a fourfold likelihood of GD clinical diagnoses in autistic compared to non-autistic children (i.e., 0.07% of autistic children and 0.01% of non-autistic children)[17]. Despite the differences in percentages of transgender and gender-diverse identities in the studies using CBCL and clinical GD information, the relative rates are largely similar (between 5.7 and 7.7). A second set of studies has investigated rates of autism in both children and adolescents[21–23] and adults[24,25] with GD. These studies have identified that between 4.8% and 26% of individuals who present at GD clinics have an autism diagnosis based on several different criteria. The largest of these studies ($N = 532$[24], and $N = 540$[25]) identified that 6.0% and 4.8% respectively of these individuals are autistic, based on review of clinical and medical records. Although none of these studies have used a matched control sample to investigate the relative rates of autism diagnoses, using a baseline population estimate of 1–2%[2–8] suggests that autism diagnoses are significantly elevated in individuals presenting at GD clinics. A third group of studies have identified elevated traits related to autism in individuals with gender diversity[24,26–34] compared to cisgender individuals. These studies have not investigated whether atypical sensory sensitivity (now

defined as a core feature of autism[1]) is elevated in transgender and gender-diverse individuals.

The existing literature is heterogeneous, conducted using different methods, across age ranges and nationalities. These studies demonstrate an increased occurrence of autism in gender-diverse individuals or individuals from GD clinics. However, almost all studies were conducted using modest sample sizes (a typical sample size is in a few hundreds). Whilst these have the advantage of carefully characterizing gender identity, they may not correctly estimate the effect sizes as the Odds Ratios (ORs) may be biased away from zero[35,36]. Larger samples would minimize the bias, but a bias will likely exist in most samples. Additionally, most studies have focused on individuals from GD clinics. However, not all transgender and gender-diverse individuals have GD, and the rates of autism in GD individuals may be different from rates of autism in transgender and gender-diverse individuals. It is also likely that young people attending GD clinics represent young people with the most intense gender dysphoria, such that it warrants a referral for clinical care, and/or those young people who can access this care (e.g., with parents who are more tolerant of difference, or who have greater resources, etc.). Therefore, it is important to understand what the odds are of being diagnosed as autistic in transgender and gender-diverse individuals at large, not solely in those recruited through GD clinics.

In parallel, studies have also investigated the rates of mental health conditions and mental distress in transgender and gender-diverse individuals, including individuals with GD (e.g., references[37–44]). The literature is heterogeneous with varying research methodologies and sample sizes[45]. Two recent reviews identify higher rates of mental health conditions and mental distress (notably depression, anxiety, and substance use disorders) in transgender and gender-diverse individuals compared to cisgender individuals[40,45]. Most of this research has focused on depression, substance misuse, and anxiety, with limited research on neurodevelopmental and other psychiatric conditions. It is unclear how the elevated rates of autism diagnosis in transgender and gender-diverse individuals compare to other neurodevelopmental and psychiatric conditions. To our knowledge, barring one study[16], none of the existing studies of autism and gender identity have compared the rates of other related neurodevelopmental and psychiatric conditions in transgender and gender-diverse individuals versus cisgender individuals, making it difficult to estimate if the observed effects are specific to autism.

The availability of large datasets to investigate the link between autism and gender identity is currently limited to internet-based surveys. As far as we are aware, there is no large-scale national or regional registry with information available on both gender identity[40] (not limited to individuals with gender dysphoria) and autism diagnosis. We address these issues using four large-scale cross-sectional, internet-based datasets, and one longitudinal dataset, all sampled using a convenience framework. Using these five datasets, we investigate if transgender and gender-diverse individuals, compared to cisgender individuals, have: (1) elevated rates of autism diagnosis; (2) elevated autistic traits, systemizing traits, sensory hypersensitivity traits, and reduced empathy traits, all related to autism; and (3) elevated rates of any of six neurodevelopmental and psychiatric conditions that commonly co-occur with autism (attention-deficit/hyperactivity disorder (ADHD), major depressive disorder (depression), bipolar disorder, obsessive-compulsive disorder (OCD), learning disorder (also known as specific learning disorder), and schizophrenia)[46,47] (Fig. 1). Finally, whilst the previous literature has provided compelling evidence that autism is under-diagnosed (or mis-diagnosed as other conditions) in cisgender females, it is unclear if this is true of transgender and gender-diverse individuals[48–50]. So, as an

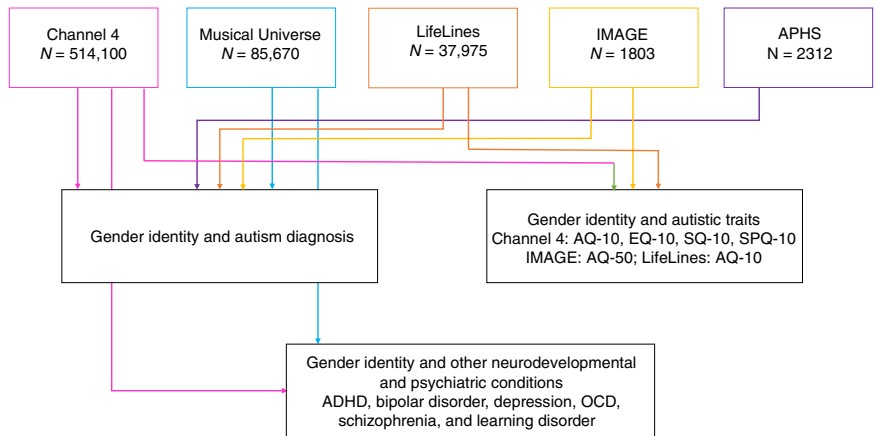

**Fig. 1 Schematic diagram of the study.** This figure provides a schematic overview of the study. In this study we investigated three questions, presented in the red boxes. For each question, the primary dataset was the Channel 4 dataset (pink box). We used four validation datasets to validate the results—Musical Universe (cyan box), LifeLines (orange box), IMAGE (yellow box), and APHS (purple box). Colored arrows from the dataset boxes to the questions indicate which questions were investigated in which datasets. AQ-10 (Autism Spectrum Quotient-10), SQ-10 (Systemizing Quotient-10), EQ-10 (Empathy Quotient-10), SPQ-10 (Sensory Perception Quotient-10), AQ-50 (Autism Spectrum Quotient-50), ADHD (Attention-Deficit/Hyperactivity Disorder), OCD (Obsessive-Compulsive Disorder).

exploratory analysis, we investigate whether transgender and gender-diverse individuals are more likely to suspect that they have undiagnosed autism compared to cisgender individuals.

## Results

**Rates of autism diagnosis.** We first investigated whether rates of autism diagnosis differed by gender in the C4 dataset. A $\chi^2$ test identified a significant difference in autism diagnosis based on gender ($\chi^2$ (2) = 3316, $\varphi$ = 0.08, $p$ value < $2 \times 10^{-16}$). Transgender and gender-diverse individuals had higher rates of autism diagnosis compared to cisgender males (OR = 4.21, 95% CI = 3.85–4.60, $p$ value < $2 \times 10^{-16}$), cisgender females (OR = 6.80, 95%CI = 6.22–7.42, $p$ value < $2 \times 10^{-16}$), and cisgender individuals altogether (i.e., cisgender males and cisgender females combined) (OR = 5.53, 95%CI = 5.06–6.04, $p$ value < $2 \times 10^{-16}$) (Fig. 2). After accounting for age and educational attainment, transgender and gender-diverse individuals had higher rates of autism diagnosis compared to cisgender males (OR = 3.88, 95%CI = 3.54–4.25, $p$ value < $2 \times 10^{-16}$), cisgender females (OR = 5.31, 95%CI = 4.85–5.82, $p$ value < $2 \times 10^{-16}$), and cisgender individuals altogether (OR = 4.59, 95%CI = 4.20–5.03, $p$ value < $2 \times 10^{-16}$) (Fig. 2).

Given the limitations of the C4 dataset, we investigated this hypothesis in four other independently recruited datasets: MU, IMAGE, APHS, and LifeLines ("Methods"). $\chi^2$ tests identified significant gender-based differences in autism diagnosis rates ($p$ value < $1 \times 10^{-5}$ in all datasets). Transgender and gender-diverse individuals had higher rates of autism diagnosis compared to cisgender males (MU: OR = 5.5, 95%CI = 4.10–7.28, $p$ value < $2 \times 10^{-16}$; IMAGE: OR = 6.36, 95%CI = 3.75–10.93, $p$ value = $6.32 \times 10^{-14}$; APHS: OR = 4.46, 95%CI = 2.95–6.96, $p$ value = $3.6 \times 10^{-13}$; LifeLines: OR = 3.63, 95%CI = 1.12–11.73, $p$ value = 0.02), cisgender females (MU: OR = 9.92, 95%CI = 7.32–13.20, $p$ value < $2 \times 10^{-16}$; IMAGE: OR = 5.35, 95%CI = 3.14–9.24, $p$ value = $5.23 \times 10^{-11}$; APHS: OR = 6.66, 95%CI = 4.45–10.29, $p$ value < $2 \times 10^{-16}$; Life-Lines: OR = 6.88, 95%CI = 2.27–20.85, $p$ value = $1 \times 10^{-4}$), and cisgender individuals altogether (MU: OR = 7.08, 95%CI = 5.28–9.30, $p$ value < $2 \times 10^{-16}$; IMAGE: OR = 5.90, 95%CI = 3.52–10.02, $p$ value = $1.80 \times 10^{-13}$; APHS: OR = 5.77, 95%CI = 3.88–8.86, $p$ value < $2 \times 10^{-16}$; LifeLines: OR = 5.50, 95% CI = 1.60–16.60, $p$ value = 0.002). These results were statistically

significant after accounting for age and educational attainment in three of the four cohorts (transgender and gender-diverse vs. cisgender: MU: OR = 6.07, 95%CI = 4.56–8.08, $p$ value < $2 \times 10^{-16}$; IMAGE: OR = 6.36, 95% CI = 3.34–12.13, $p$ value = $1.08 \times 10^{-9}$; APHS: OR = 6.28, 95%CI = 4.13–9.53, $p$ value < $2 \times 10^{-16}$). In addition, we identified concordant effect direction in the LifeLines cohort (LifeLines: OR = 3.03, 95% CI = 0.72–12.76, $p$ value = 0.13), though this was not statistically significant due to the low statistical power (Supplementary Note). Supplementary Table S3 provides the results for all three genders.

Additional sensitivity analysis in the MU dataset conducted by separating the cisgender group into cisgender males and cisgender females and the transgender and gender-diverse group into "transgender" and "other" indicated that both the non-cisgender groups had higher rates of autism diagnosis compared to both cisgender males and cisgender females (Supplementary Table S4).

Given that we did not collect information on sex and gender separately in the MU and the C4 datasets, we further investigated if the adjusted ORs (transgender and gender-diverse vs. cisgender) were significantly different for the APHS, IMAGE, and LifeLines datasets when compared to the MU and the C4 datasets. We used a subsampling bootstrap approach (10,000 sub-samples) to test this and calculated empirical p-values ("Methods"). Empirical $p$ values suggested that the ORs for the APHS ($p$ value = 0.078), IMAGE ($p$ value = 0.11), and LifeLines ($p$ value = 0.84) datasets were not statistically different from the ORs observed in the 10,000 samples generated from the C4 dataset. Similarly, empirical $p$ values for the APHS ($p$ value = 0.56), IMAGE ($p$ value = 0.44), and LifeLines ($p$ value = 0.85) datasets suggested that the ORs were not statistically different from that observed in the 10,000 permuted samples generated from the MU dataset.

We also investigated if rates of transgender and gender diversity are higher in individuals diagnosed with autism using a logistic regression framework after accounting for age and educational attainment. We identified significant associations in four of the five dataset (C4: OR = 4.66, 95%CI = 4.26–5.10, $p$ value < $2 \times 10^{-16}$; MU: OR = 6.05, 95%CI = 4.55–8.05, $p$ value < $2 \times 10^{-16}$; IMAGE: OR = 6.35, 95%CI = 3.32–12.11, $p$ value = $2.1 \times 10^{-8}$; APHS: OR = 6.31, 95%CI = 4.14–9.62, $p$ value < $2 \times 10^{-16}$) and a

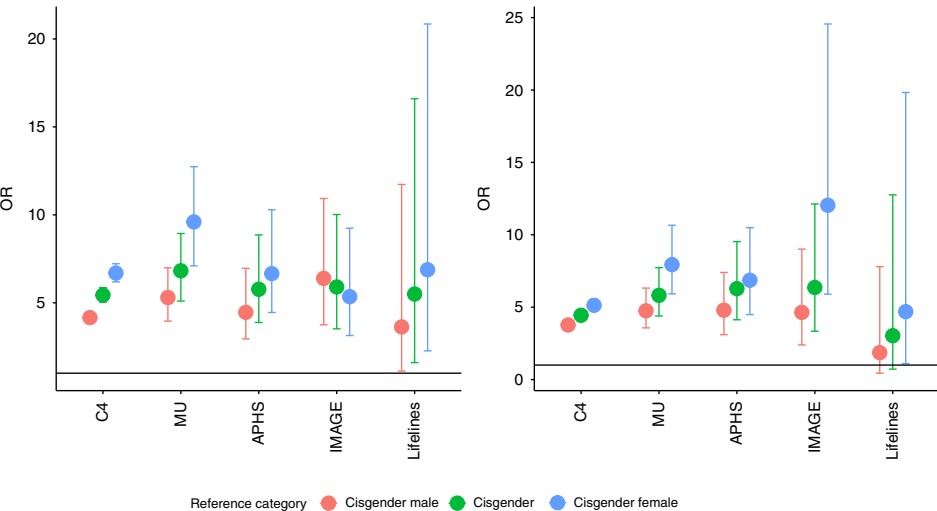

**Fig. 2 ORs and 95% CIs for autism in transgender and gender-diverse individuals compared to cisgender males, cisgender females, and cisgender individuals altogether. a** This figure provides the unadjusted Odds Ratios (ORs, point) and 95% CIs for autism in transgender and gender-diverse individuals compared to either cisgender males, cisgender females, or cisgender (cisgender males and cisgender females) individuals in five datasets (C4: $N = 514,100$; MU: $N = 85,670$; APHS: $N = 2312$; IMAGE: $N = 1803$; and LifeLines: $N = 37,975$). **b** This figure provides adjusted ORs (point) and 95% CIs for autism in transgender and gender-diverse individuals compared to cisgender males, cisgender females, or all cisgender individuals in five datasets (C4: $N = 514,100$; MU: $N = 85,670$; APHS: $N = 2312$; IMAGE: $N = 1803$; and LifeLines: $N = 37,975$). ORs have been adjusted for age, educational attainment, and in the case of IMAGE dataset, an additional dummy variable for study (see "Supplementary Methods"). The y-axis is on the same scale for both the panels. The differences in ORs for the IMAGE dataset between Models 1 and 2 is primarily due to the inclusion of "study" group as a covariate. Specifically, the IMAGE dataset consists of individuals recruited into a study of mathematics and autism ("Methods"). Whilst the mathematics group is predominantly male and have higher educational attainment (all have at least an undergraduate degree), the case–control group had a more balanced ratio and a wider range of educational attainment. Covarying for the study the participants have been recruited into (mathematics or autism case–control) changes the ORs.

concordant effect direction in the LifeLines dataset (OR = 2.91, 95% CI = 0.69–12.20, $p$ value = 0.14).

**Traits related to autism.** As seen in cisgender individuals[51], autistic transgender and gender-diverse individuals scored higher on the AQ-10, SQ-10, and SPQ-10, and lower on the EQ-10 compared to non-autistic transgender and gender-diverse individuals (Cohen's D: 0.54−0.72, $p$-value $< 2 \times 10^{-16}$, Supplementary Tables S5 and S6).

We next investigated gender differences in scores on the AQ-10, SQ-10, EQ-10, and SPQ-10 in autistic and non-autistic individuals separately in the C4 dataset. In both autistic and non-autistic individuals separately, ANOVA identified significant differences based on gender on all four measures ($p$ value $< 2 \times 10^{-16}$ in all comparisons). Post-hoc t-tests indicated significant differences between groups across all measures: transgender and gender-diverse individuals scored higher on the AQ-10, SQ-10, and SPQ-10, and lower on the EQ-10 compared to both cisgender males and cisgender females. The effect sizes for differences in scores were larger for the cisgender male vs. transgender and gender-diverse as well as cisgender female vs. transgender and gender-diverse tests compared to the cisgender male vs. cisgender female tests across all four measures in both non-autistic and autistic individuals (Supplementary Tables S5 and S6).

For both cisgender male vs. transgender and gender-diverse as well as cisgender female vs. transgender and gender-diverse comparisons, effect sizes were larger in autistic individuals (Cohen's D: 0.55–1.05) compared to the same analyses in non-autistic individuals (Cohen's D: 0.32–0.96). This contrasts with cisgender male vs. cisgender female gender differences for these measures, which are attenuated in autistic individuals compared to non-autistic individuals (Supplementary Tables S5 and S6 and Fig. 3).

We repeated the analyses after accounting for autism diagnosis, age, and educational attainment. Transgender and gender-diverse individuals scored higher ($p$ value $< 2 \times 10^{-16}$ for all) than both cisgender males and cisgender females on the AQ-10 (cisgender males: Beta = 0.89 ± 0.02, cisgender females: Beta = 1.05 ± 0.02), the SQ-10 (cisgender males: Beta = 0.66 ± 0.02, cisgender females: Beta = 0.99 ± 0.02), and the SPQ-10 (cisgender males: Beta = 0.66 ± 0.02, cisgender females: Beta = 0.55 ± 0.02), and lower on the EQ-10 (cisgender males: Beta = −0.33 ± 0.02, cisgender females: Beta = −0.70 ± 0.02) (Fig. 3 and Supplementary Fig. S1). We replicated this in two datasets: the IMAGE dataset using the AQ-50 and the LifeLines dataset using the AQ-10. In the IMAGE dataset, transgender and gender-diverse individuals scored higher than both cisgender males (Beta = 0.45 ± 0.11, $p$ value = $3.09 \times 10^{-5}$) and cisgender females (0.52 ± 0.11, $p$ value $< 1.80 \times 10^{-6}$). In the LifeLines dataset, transgender and gender-diverse individuals scored higher than cisgender females (Beta = 1.23 ± 0.25, $p$ value = $1.4 \times 10^{-6}$) and nominally higher than cisgender males (Beta = 0.51 ± 0.25, $p$ value = 0.045).

The previous analyses investigated the association between gender identity and traits related to autism individually. We next investigated if there are differences in the standardized discrepancy between the EQ-10 and the SQ-10 in the three gender categories using "Brain Types". Compared to both non-autistic cisgender males and non-autistic cisgender females, non-autistic transgender and gender-diverse individuals were significantly more likely to be classified as Type S (cisgender males 40.23%, cisgender females 25.58%, transgender and gender-diverse 53%) or Extreme Type S (cisgender males 4.14%, cisgender females 1.69%, transgender and gender-diverse 13.15%) ($p$ value $< 2 \times 10^{-16}$). This was more pronounced in autistic transgender and gender-diverse individuals compared to autistic cisgender individuals (Extreme Type S: cisgender males 11.42%, cisgender females 7.55%, and transgender and gender-

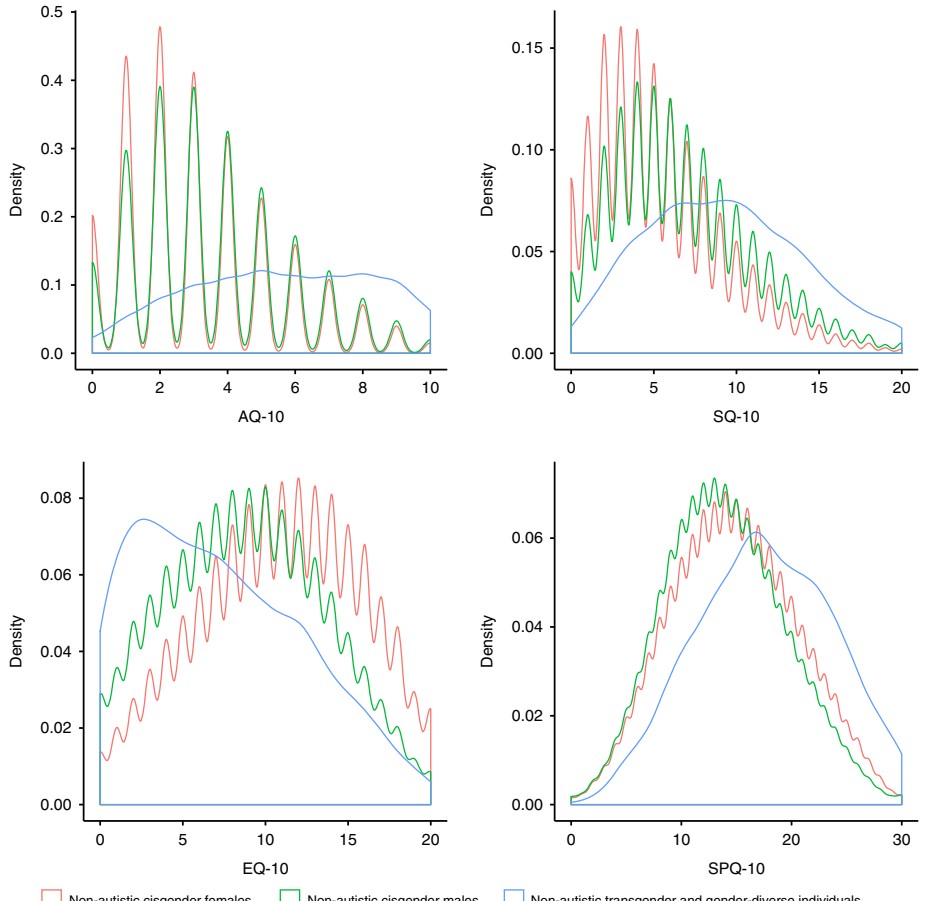

**Fig. 3 Kernel density plot of scores on the four self-report measures in the C4 Dataset for non-autistic individuals only.** This figure provides kernel density plots for scores on the four self-report measures (AQ-10, EQ-10, SQ-10, and SPQ-10) for non-autistic participants from the C4 dataset ($N = 514,100$) based on their gender (cisgender males, cisgender females, transgender and gender-diverse individuals). Scales on the axes are different between the panels. See Supplementary Fig. S1 which provides kernel density plots for all four measures for both autistic and non-autistic individuals. The non-autistic transgender and gender-diverse kernel density plots appear smoother due to the relatively low number of participants included, hence providing less resolution in the kernel density estimates when compared to the non-autistic cisgender males and non-autistic kernel density plots.

diverse 34.73%; Type S: cisgender males 50.97%, cisgender females 42.29%, transgender and gender-diverse 51.79%) ($p$ value $< 2 \times 10^{-16}$). (Supplementary Table S7 and Supplementary Fig. S2). Cumulatively, in autistic individuals, 86.52% of transgender and gender-diverse individuals were classified as Type S or Extreme Types S compared to 62.39% of cisgender males. In both autistic and non-autistic transgender and gender-diverse individuals, observed values were significantly shifted towards Type S and Extreme Type S compared to what is expected ($p$ value $< 2 \times 10^{-16}$).

**Rates of other neurodevelopmental and psychiatric conditions.** We next investigated if rates of six other neurodevelopmental and psychiatric conditions (ADHD, bipolar disorder, depression, learning disorder, OCD, and schizophrenia) differed by gender in the C4 dataset. Compared to cisgender individuals, transgender and gender-diverse individuals had elevated rates of all these conditions, with the highest effect size for schizophrenia (OR = 28.52, 95% CI = 24.17–33.66, $p$ value $< 2 \times 10^{-16}$) and the lowest for learning disorders (OR = 3.48, 95%CI = 3.09–3.91, $p$ value $< 2 \times 10^{-16}$) (Supplementary Table S8). Including age and educational attainment as covariates (Model 2) attenuated the ORs only modestly (ORs: 3.08 (learning disorders) to 19.73 (schizophrenia)). However, the ORs were substantially attenuated when autistic individuals

were excluded, i.e., Model 3 (1.92 (learning disorders) to 6.39 (schizophrenia)) (Supplementary Table S8). Notably, there was a considerable attenuation in the OR for schizophrenia. The ORs for autism, ADHD, bipolar disorder and depression were similar to each other. In comparison, the ORs for OCD and LD were about half that for autism. Supplementary Table S9 provides results of the analyses repeated for the three genders (cisgender male, cisgender female, and transgender and gender-diverse).

We repeated the analyses for five of the six conditions tested above in the MU dataset. Compared to cisgender individuals, transgender and gender-diverse individuals reported higher rates of all five conditions (Model 1; OR: 2.15 (schizophrenia) to 3.83 (depression)), with the results for schizophrenia not being statistically significant, possibly due to small sample size (Fig. 4). These results were similar after accounting for educational attainment and age (Model 2; OR: 1.81 (schizophrenia) to 3.89 (depression)), and additionally, after excluding autistic individuals (Model 3 OR: 1.11 (schizophrenia) to 3.91 (depression)) (Supplementary Table S8). In contrast to the C4 dataset, in the MU dataset, the ORs for autism was the largest, followed by the two mood disorders (depression and bipolar disorder). Notably, the OR for depression was similar in both the C4 and the MU datasets. Supplementary Table S9 provides results of the analyses repeated for three genders (cisgender male, cisgender female, and transgender and gender-diverse).

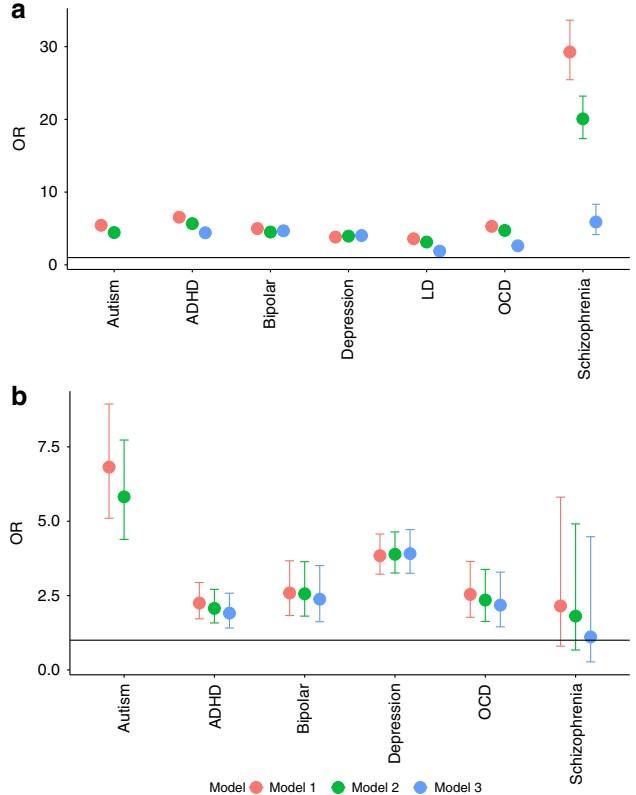

**Fig. 4 ORs and 95% CIs for other neurodevelopmental and psychiatric conditions in transgender and gender-diverse individuals compared to cisgender individuals. a** This figure provides the Odds Ratios (ORs, point) and 95% CIs for diagnosis of autism and six other neurodevelopmental and psychiatric conditions in transgender and gender-diverse individuals compared to cisgender individuals in the C4 dataset ($N = 514,100$). We did not employ Model 3 for autism as it was conducted after excluding autistic individuals in the dataset. ORs have been calculated using three models (see Methods). ADHD = Attention-Deficit/Hyperactivity Disorder; OCD = Obsessive-Compulsive Disorder; LD = Learning Disorder. **b** This figure provides the same, but for the MU dataset ($N = 85,670$). Information on LD was not available in the MU dataset. The y-axis is on a different scale from the panel above.

To further clarify the role of autism compared to other neurodevelopmental and psychiatric conditions, we conducted multiple regressions to investigate the relative effects of association of autism on transgender and gender-diverse identities compared to other neurodevelopmental and psychiatric conditions. In the C4 dataset, depression had the highest OR (OR = 3.55, 95%CI = 3.84–3.29, $p$ value < $2 \times 10^{-16}$) followed by autism (OR = 3.43, 95%CI = 3.79–3.11, $p$ value < $2 \times 10^{-16}$). In the MU dataset, we obtained very similar ORs. Autism had the highest OR (OR = 3.94, 95%CI = 5.61–2.77, $p$ value < $2 \times 10^{-16}$) followed by depression (OR = 3.50, 95%CI = 4.25–2.89, $p$ value < $2 \times 10^{-16}$). ORs for other conditions are provided in the Supplementary Table S10.

**Exploratory analysis: rates of suspected autism.** In the IMAGE dataset, we also investigated if transgender and gender-diverse individuals were more likely to suspect they had undiagnosed autism compared to cisgender individuals. A $\chi^2$ test identified a significant difference between genders ($\chi^2$ (2) = 42.087, $\varphi = 0.15$, $p$ value = $7.52 \times 10^{-10}$). Transgender and gender-diverse individuals were more likely to suspect they had undiagnosed autism compared to cisgender males (OR = 4.32, 95%CI = 1.94–10.10,

$p$ value = $2.51 \times 10^{-4}$), cisgender females (OR = 7.99, 95% CI = 3.54–18.92, $p$ value = $3.13 \times 10^{-8}$), and cisgender male and female individuals altogether (OR = 5.47, 95%CI = 2.47–12.72, $p$ value = $9.01 \times 10^{-6}$).

## Discussion

In this study, we investigated three primary questions, and an additional exploratory question using five different, large-scale datasets. First, across all five datasets, transgender and gender-diverse individuals were 3.03 to 6.36 times as likely to be autistic than were cisgender individuals, after controlling for age and educational attainment. Second, transgender and gender-diverse individuals scored significantly higher on self-report measures of autistic traits, systemizing and sensory sensitivity and scored significantly lower on empathy traits compared to cisgender individuals. Third, in two datasets with available data, transgender and gender-diverse individuals had elevated rates of multiple other neurodevelopmental and psychiatric conditions. Finally, exploratory analysis identified that transgender and gender-diverse individuals were more likely to report that they suspected they had undiagnosed autism.

These associations between gender identity and autism diagnoses are unlikely to be false positives for multiple reasons. First, we observe consistent effect directions across multiple datasets with very different recruitment strategies, ascertainment biases, cultural backgrounds, and age ranges. The effects after accounting for age and educational attainment were statistically significant for four of the five datasets, and in the same direction for the fifth (i.e., LifeLines cohort). The lack of statistical significance is due to the low statistical power of the LifeLines dataset, because participants were older and healthier as individuals with severe mental health conditions were excluded at the time of recruitment, and individuals with higher genetic likelihood for mental health conditions are likely to drop out from longitudinal studies[52,53]. Second, comparing the ORs of the three smaller samples (IMAGE, APHS, and LifeLines) to bootstrapped ORs from 10,000 subsamples in the two largest samples (C4 and MU) did not identify statistically significant differences in ORs. This indicates that the ORs are similar regardless of different recruitment strategies and different methods used to ascertain gender and autism. Third, sensitivity analysis in the MU dataset did not identify differences in the rates of autism diagnosis between participants who indicated "Other" vs. "Transgender". Fourth, the ORs observed in this study are similar to those observed in participants from GD clinics[17], suggesting that ORs observed using an internet-based convenience sampling framework is similar to ORs observed in GD clinic-based samples.

Supporting the association between gender identity and autism diagnoses, transgender and gender-diverse individuals also had higher scores on self-report measures of autistic traits, sensory sensitivity, and systemizing, and lower scores on a self-report measure of empathy traits, compared to cisgender individuals. The transgender and gender-diverse vs. cisgender effect sizes are equivalent to or larger than the autism vs. non-autism effect sizes and the cisgender male vs. cisgender female effect sizes in non-autistic individuals. Importantly, these effects were also observed when investigating the discrepancy of scores on the EQ-10 and SQ-10 using the "Brain Types" analyses. In addition, in a relatively smaller sample (IMAGE), transgender and gender-diverse individuals were more likely to suspect they had undiagnosed autism. Taken together, our analyses indicate that transgender and gender-diverse individuals are more likely to be autistic compared to cisgender individuals, and further that undiagnosed autism may also be higher in transgender and gender-diverse individuals.

However, this association with gender identity is not specific to autism. In two datasets, transgender and gender-diverse individuals also had elevated rates of ADHD, bipolar disorder, depression, OCD, learning disorders, and schizophrenia compared to cisgender individuals. In one of the two datasets, we tested and confirmed that transgender and gender-diverse individuals had higher rates of learning disorders compared to cisgender individuals. In the C4 dataset, we identified elevated rates of schizophrenia in transgender and gender-diverse individuals compared to cisgender individuals but were unable to replicate this in the MU dataset.

Our multiple regression analyses helped clarify the relative association strengths of these conditions with transgender and gender-diverse individuals. In both the MU and the C4 datasets, autism and depression had the highest effect sizes. Notably, in the MU dataset, none of the other conditions were significantly elevated in transgender and gender-diverse individuals after controlling for autism and depression, which is discordant with the results identified in the C4 datasets. This discrepancy in the results may be due to differences in sample sizes, ascertainment, or other cohort characteristics. For instance, the C4 study directly recruited participants to an autism study. This may oversample individuals with other co-occurring mental health conditions. In contrast, the MU dataset is a convenience sample collected over many months. There is some evidence to suggest that individuals with elevated genetic liability for schizophrenia, ADHD, and depression may be less likely to participate in studies[52,53], and, as a result, they may be underrepresented in the MU dataset. In addition, most of the participants in the C4 are from the UK, whilst most of the MU participants are from the US. Differences in diagnostic practices may also contribute to sampling differences. A more comprehensive investigation of the relative rates of neurodevelopmental and psychiatric conditions in transgender and gender-diverse individuals compared to cisgender individuals is warranted.

The elevated rates of autism and other conditions must be considered against other hypotheses that may explain the observed results due to the non-probabilistic nature of the sample. Specifically, for autism, one alternative hypothesis is that transgender and gender-diverse individuals may be more likely to report higher rates of autistic traits due to long-standing experiences and feelings of "not fitting in socially", with true levels of autistic traits being comparable between cisgender and transgender and gender-diverse individuals. Although this is possible, other studies have reported elevated autistic traits measured using parent- or teacher-report instruments in individuals with GD[31,33]. Importantly, in our study, we note that the shift in scores in transgender and gender-diverse individuals is observed across both social (EQ-10) and non-social (SPQ-10 and SQ-10) measures of traits related to autism, which themselves are only partly correlated[51,54,55]. Notably, transgender and gender-diverse individuals also score higher on the SPQ-10, a measure of sensory sensitivity, and response to items on this measure are unlikely to be influenced by social gender norms.

Another alternative hypothesis is that autistic transgender and gender-diverse individuals may be more likely to participate in these studies compared to autistic cisgender individuals (i.e., selection bias). However, this is unlikely: the datasets were not collected to specifically investigate the links between gender and rates of autism diagnosis. Whilst autistic individuals may be more likely to participate in the autism-related studies (C4, APHS, and IMAGE), it is unlikely that this will be biased towards autistic transgender and gender-diverse compared to autistic cisgender individuals. In addition, two of the datasets (MU and LifeLines) were not collected specifically for an autism-based study. Further, the LifeLines also has a healthy volunteer bias, which is likely to

attenuate ORs. In other words, a strength of this study is that none of the datasets were collected to specifically test the association between autism and gender identity. Furthermore, similar ORs have been observed in a large-scale study of autism in participants of GD clinics which are unlikely to be affected by this specific type of selection bias[17], providing further corroboration to our findings.

Whilst our study does not test causality, a few hypotheses may explain the over-representation of autism and other neurodevelopmental and psychiatric conditions in transgender and gender-diverse individuals. First, autistic individuals may conform less to societal norms compared to non-autistic individuals, which may partly explain why a greater number of autistic individuals identify outside the stereotypical gender binary. Second, prenatal mechanisms (e.g., sex steroid hormones) shaping brain development have been shown to contribute to both autism (and associated neurodevelopmental conditions) and gender role behavior[56–60]. It is unclear if prenatal sex steroid hormones also contribute to gender identity and this should be investigated in future studies. Neurodevelopmental conditions such as ADHD and learning disorders frequently co-occur with autism[47], and genetic evidence suggests a shared underlying liability for many of the co-occurring neurodevelopmental and psychiatric conditions[61,62]. Finally, an alternative but not mutually exclusive explanation is that transgender and gender-diverse individuals have elevated vulnerabilities for multiple psychiatric challenges related to stressful life experiences in the contexts of unfriendly environments, discrimination, abuse and victimization, explaining the elevated rates of mental health diagnoses[63,64].

These findings must be interpreted in light of the lived experiences, rights, and clinical and daily life needs of transgender and gender-diverse individuals. Both autistic individuals and transgender and gender-diverse individuals are marginalized groups where the currently available support and understanding is inadequate[65]. Both groups are also more likely than others to engage in self-harm, suicidal ideation and suicidal behaviors, and to have other vulnerabilities[63,66–68]. This intersection of autism and gender diversity can be doubly distressing if adequate safeguarding and support are not provided. A recent study demonstrated that a third of autistic individuals had their gender identity questioned because they were autistic[65]. There is a need to ensure that autistic transgender and gender-diverse individuals have the right to express their gender, live with dignity, and receive social and legal recognition of their gender[69] (also see: https://autisticadvocacy.org/wp-content/uploads/2016/06/joint_statement_trans_autistic_GNC_people.pdf). Additionally, recent studies demonstrate that autistic characteristics partly differ between cisgender males and cisgender females[50,70,71]. However, it is still unclear if autistic characteristics differ in transgender and gender-diverse individuals compared to cisgender individuals. This co-occurrence requires gender-informed and neurodiversity-informed clinical care for autistic transgender and gender-diverse individuals.

There are caveats to this study. First, in two of the datasets we excluded intersex individuals, but this was not an option in other datasets (C4, LifeLines and MU). Second, there is a possibility that some nonbinary, gender-neutral, or other gender-diverse individuals may not identify with the "transgender" term in the C4 dataset as we did not concurrently provide the "transgender" and "other" options. Third, some gender-aware individuals may respond by providing their sex rather than their gender. It is difficult to disentangle this. However, the magnitude of the sample size suggests that the effects of such misclassification will have a minimal effect on the analyzes and findings. Supporting this, subsampling bootstrap analyses indicate that the ORs are similar across the different datasets. Additionally, the ORs are

similar between the five internet-based datasets in this study and a study based on GD-clinic based samples[17]. This similarity suggests that regardless of recruitment (internet-based vs. clinic-based) or ascertainment criteria (self-report gender identity vs. clinically ascertained gender dysphoria) or age (adults vs. children), the results converge on similar ORs. Fourth, individuals with severe mental health conditions and intellectual disability are less likely to participate. Finally, these datasets are not statistically well-powered to investigate rates of autism diagnosis in transgender and gender-diverse individuals after stratifying by sex assigned at birth; thus, we have not investigated this.

In conclusion, our study demonstrates that transgender and gender-diverse individuals have elevated rates of autism diagnosis, related neurodevelopmental and psychiatric conditions, and autistic traits compared to cisgender individuals. This study has clinical implications by highlighting that we need to improve access to care and tailored support for this under-served population.

## Methods

**Overview of the datasets**. We used five datasets for this study. The largest of these (Channel 4 dataset, C4) consists of $N = 514,100$ individuals who completed online questionnaires as a part of a UK Channel 4 television program about autism. These participants self-reported their autism diagnosis, and indicated their gender based on three options "Male", "Female", and "Transgender". To address autism-related self-selection bias in this dataset, we used a second dataset (Musical Universe, MU, $N = 85,670$) recruited through a website for research about musical behavior, personality and cognition. Participants completed information about their autism diagnosis and selected their gender from four options: "Male", "Female", "Transgender" and "Other". However, neither of these two datasets have separately recorded information on sex at birth and gender, and in both datasets, participants were asked to choose their "Sex", although we acknowledge that the information collected is primarily of gender. To address this, we used two additional datasets where information was collected separately for sex at birth and gender. In the third dataset (APHS, $N = 2312$), participants were recruited for an internet-based physical health survey. Participants completed information on their autism diagnosis including when they were diagnosed and who diagnosed them, their sex at birth, and their current gender identity. The fourth dataset (IMAGE, $N = 1803$) consists of participants who were recruited for a genetic study of autism and mathematical ability. Participants completed information on their autism diagnosis, their sex at birth, and their gender. In addition, all autistic participants provided a copy of their diagnostic report to verify their diagnosis. The fifth and final dataset consists of a subset of participants from the LifeLines Cohort and Biobank[72] ($N = 37,975$) who provided information on sex assigned at birth and gender, autism diagnosis, and completed a measure of autistic traits. This dataset consists of individuals who are considerably older than those in the other four datasets, and who were recruited primarily through GP clinics. None of the five datasets were recruited specifically to investigate the association between gender diversity and autism, which limits gender-based self-selection bias.

**Channel 4 dataset: overview**. The Channel 4 dataset (C4 dataset) comprises participants who completed self-report measures as a part of the Channel 4 documentary titled "Are you autistic?", in Spring 2017[51]. A mobile-friendly website was developed and advertised on the Channel 4 TV website (https://www.channel4.com/). Participants indicated if their results could be used for research purposes. A total of 758,916 entries were recorded. Participants provided information on demographics (gender (see below for details), age, educational attainment, geographical region, handedness, occupation, autism and other neurodevelopmental or psychiatric diagnosis) and completed four self-report measures. Participants who consented to share their data for research were asked: "Have you taken this survey before? To make sure our data are as accurate and as useful as possible please tell us if you've taken this survey before." If participants indicated that they had taken the survey before, they were marked as duplicates. After removing duplicates, we were left with a total of 695,166 participants. We were unable to use IP addresses to identify duplicates due to ethical constraints. We included participants aged 15 to 90 years, in line with previous research[51]. Participants were asked to indicate their "Sex" using one of four options: "Male", "Female", "Transgender" and "Prefer not to say". Whilst "Sex" was asked in the survey, we recognize that the information provided here is of sex or gender, or both and we refer to this as gender throughout the manuscript. Whilst designing the survey we did not make a distinction between gender and sex as these terms are often used interchangeably in the general population. We further removed individuals who did not provide information on gender ("Prefer not to say"), resulting in $N = 675,360$ individuals.

**Channel 4: ascertaining gender identity**. During data collection, information on gender was initially collected using four options listed above. However, towards the end of the data collection phase, the "Transgender" option was modified to "Other" to make it more inclusive. For this study, we restricted our analysis to only those participants from the first phase of data collection who could choose from "Male", "Female", "Transgender" and "Prefer not to say", as this makes it clearer for interpreting the data. This resulted in 514,100 individuals whose gender was either "Male" ($N = 193,398$), "Female" ($N = 317,891$), or "Transgender" ($N = 2811$ or 0.55%).

**Channel 4: ascertaining diagnosis of autism and other conditions**. 27,919 participants (5.4%) indicated they had an autism diagnosis (cisgender males = 13,317; cisgender females = 13,934, transgender and gender-diverse = 668). Diagnoses of autism and other psychiatric conditions were asked using the question: "Have you been formally diagnosed with any of the following (please click all that apply?)". For other psychiatric conditions, participants could choose from ADHD, bipolar disorder, depression, learning disorder, schizophrenia, and OCD. The wording of the question should typically preclude (though not completely eliminate) self-diagnosed individuals. Participants indicated they had the following diagnoses: ADHD ($N = 19,300$), bipolar disorder ($N = 9025$), depression ($N = 122,829$), learning disorder ($N = 18,559$), OCD ($N = 13,115$), and schizophrenia ($N = 1321$). These were not mutually exclusive, as individuals could endorse several diagnoses. In addition, participants provided information on their educational attainment and age (Supplementary Tables S1 and S2).

**Channel 4: measures of traits related to autism**. All participants completed four short, self-report psychological trait measures: the Autism Spectrum Quotient-10 (AQ-10)[73], a widely-used measure of autistic traits; the Empathy Quotient-10 (EQ-10)[51], a measure of empathy traits; the Systemizing Quotient-10 (SQ-10)[51] (10 items from the Systemizing Quotient–Revised[74], but referred to here as Systemizing Quotient-10), a measure of systemizing traits (the drive to analyze or build a system[75]); and the Sensory Perception Quotient-10 (SPQ-10)[51], a measure of sensory sensitivity. Using the SQ-10 and the EQ-10 data, we calculated "Brain Types"[51], which refer to an individual's cognitive profile based on the discrepancy of their scores on empathy and systemizing traits. Individuals may be classified into one of five different "Brain Types" based on the standardized discrepancy between their systemizing and empathy scores[51,76].

**Musical Universe dataset: overview of dataset**. The Musical Universe (MU) dataset consists of a total of 89,218 individuals who completed measures on musical behavior, personality, and cognition, in exchange for feedback about their scores at www.musicaluniverse.org. We identified duplicates first using IP addresses, and then, among individuals with identical IP addresses, using demographic variables—gender (see below for further information about this), age, educational attainment, occupation, and diagnosis. A total of 85,670 unique records were identified. Participants ranged in age from 18 to 88 years old (Supplementary Table S1).

**Musical Universe: ascertaining gender identity**. Similar to C4, the MU data collection did not make a clear distinction between gender and sex. Participants were asked for their "Sex" where they could choose one of four options: "Male" (42,291 non-autistic and 666 autistic), "Female" (41,659 non-autistic and 365 autistic), "Transgender" (361), and "Other" (328) (Supplementary Table S1). However, we recognize that participants have actually provided information on their gender and we refer to this as gender throughout the manuscript. In the primary analyses, we combined participants who chose the "Transgender" and "Other" option into the transgender and gender-diverse group (634 non-autistic and 55 autistic individuals) and conducted further sensitivity analyses using only individuals who chose the "Transgender" option. We decided to combine the two groups as some individuals who are transgender and gender-diverse in the broad sense (i.e., their gender is different from their sex assigned at birth) may not identify as transgender and may interpret the term transgender more narrowly (i.e., their binary gender identity is opposite to the binary sex assigned at birth).

**Musical Universe: ascertaining diagnosis of autism and other conditions**. Participants were asked if they had a formal diagnosis of autism from a professional. This should typically preclude (though not completely eliminate) self-diagnosed autistic individuals from participating. A total of 1,086 participants indicated that they had an autism diagnosis (Supplementary Table S1). In addition, they were asked if they had a formal diagnosis of additional mental health conditions. A subset of participants ($N = 54,127$) indicated if they had a formal diagnosis of: 1. ADHD ($N = 3189$, 5.89%); 2. Bipolar disorder ($N = 1532$, 2.83%); 3. Depression ($N = 11,919$, 22.02%); 4. OCD ($N = 1419$, 2.62%); and 5. Schizophrenia ($N = 202$, 0.37%).

**Autism Physical Health Survey: overview of dataset**. The Autism Physical Health Survey (APHS) dataset consists of 2312 individuals aged 16–90 years who were recruited via the Cambridge Autism Research Database (CARD), autism

charities and support groups, and social media as a part of a study investigating the association between autism and physical health conditions. The study employed an anonymous, online self-report survey via Qualtrics. Participants were asked questions regarding their demographics, lifestyle factors (including diet, exercise, sleep, and sexual/social history), personal medical history, and family medical history for all first-degree, biological relatives. As the study was anonymous (and we did not collect IP addresses), we excluded records that we determined were likely to be duplicates. We excluded all records that matched a previous record across 11 categories: whether or not they had an autism diagnosis, specific autism diagnosis, type of practitioner who diagnosed them, year of diagnosis, syndromic autism (if applicable), country of residence, sex assigned at birth, current gender identity, age, maternal age at birth, paternal age at birth, and educational attainment.

**Autism Physical Health Survey: ascertaining gender identity**. Participants were asked for their sex assigned at birth ("Male", "Female", "Other") and for their current gender identity ("Female" ($N = 1383$), "Male" ($N = 766$), "Non-binary" ($N = 109$), and "Other" ($N = 20$)). We removed participants who indicated "Other" for their sex assigned at birth ($N = 1$), and who did not complete information on gender identity ($N = 3$). Additionally, 33 individuals had discordant sex and gender information (7 individuals of male sex but female gender, and 26 individuals of female sex and male gender). As we did not provide a transgender option in the gender identity column, we classified these individuals as transgender. Thus, in total there were 162 individuals who were included in the transgender and gender-diverse group (Supplementary Table S1).

**Autism Physical Health Survey: ascertaining autism diagnosis**. Participants were asked to indicate if they had an autism diagnosis. Whilst we did not require participants to upload a copy of their diagnostic report, they had to provide further information about which type of clinician diagnosed them as autistic (general practitioner, neurologist, pediatrician, psychiatrist, psychologist or other (free text box)), what their specific diagnosis was, and when they were diagnosed. A total of 1082 individuals indicated that they had an autism diagnosis (Supplementary Table S1).

**The IMAGE study: overview of dataset**. The Investigating Mathematics and Autism using Genetics and Epigenetics (IMAGE) dataset consists of individuals recruited into a genetic study of autism and mathematical ability. This was done using two different research designs. The first targeted autistic and non-autistic individuals as a part of a case–control design ($N_{final} = 292$) by advertising in research databases, autism-related magazines, and on social media. The second targeted individuals who studied or were studying mathematics or a related degree ($N_{final} = 1803$) by advertising in universities, mathematics societies, in mathematics specific or alumni magazines, or on social media. Participants registered at a bespoke website and provided contact details, demographics, and completed various questionnaires. As participants provided both their names and their contact details, we used this information to remove duplicate records.

**The IMAGE study: ascertaining gender identity**. Participants were asked for their sex at birth ("Male", "Female" or "Intersex") and their gender ("Man" ($N = 994$), "Woman" ($N = 747$), "Transgender Man" ($N = 7$), "Transgender Woman" ($N = 3$), "Nonbinary" ($N = 35$), "Gender Neutral" ($N = 10$), "Other" ($N = 7$), and "Prefer not to say" ($N = 15$)). We excluded individuals who chose "Intersex" ($N = 2$) for their sex, and "Prefer not to say" ($N = 15$) for their gender. Of the remaining, we combined individuals who chose "Man" and "Woman" as the cisgender group ($N = 1741$), and the remaining into the transgender and gender-diverse group ($N = 62$). Further details are provided in Supplementary Table S1.

**The IMAGE study: ascertaining autism diagnosis**. Participants were asked if they had a diagnosis of autism on the autism spectrum (e.g., autism, Asperger Syndrome). As a part of this, we indicated that diagnosis must have been made by a qualified professional (e.g., clinical psychologist or psychiatrist). Participants were also asked when they received an autism diagnosis and who diagnosed them. In addition, autistic individuals in this study were asked to provide a copy of their diagnostic report that we used to confirm their autism diagnosis. A total of 1082 individuals indicated that they had an autism diagnosis (Supplementary Table S1). A subset of participants ($N = 1787$) provided information on educational attainment. 1417 participants indicated if they suspected they had undiagnosed autism ("Yes" or "No"). This was used to investigate if transgender and gender-diverse non-autistic individuals were more likely to suspect they had undiagnosed autism compared to non-autistic cisgender individuals.

**The IMAGE study: measures of traits related to autism**. All participants completed the AQ-50[77].

**LifeLines: overview of dataset**. The LifeLines Cohort is a Netherlands-based population cohort study, recruited between 2006 and 2013[72]. Participants were invited through their general practitioners in three northern provinces in the Netherlands (Freisland, Groningen, and Drenthe). Notably, participants were not invited if they had a severe mental health condition, which suggests that this dataset will be biased towards healthy participants. A total of 167,729 participants aged between 6 months and 93 years completed the baseline survey. The LifeLines dataset used in this study consists of 37,975 individuals from the cohort, who responded to an online questionnaire on autistic traits in summer 2019. All participants were at least 18 years of age. The participants in the LifeLines cohort were, on average, about twice as old as the participants in the C4 and the MU cohorts, and this may in part explain the relatively low number of transgender and gender-diverse individuals in this dataset. In addition, 37,574 participants provided information on their highest level of educational attainment (Supplementary Table S2).

**LifeLines: ascertaining gender identity**. Information on gender was collected using one question: "Please choose which description fits you best". This was followed by five options: "At birth I was registered as female and I am female", "At birth I was registered as male and I am male", "At birth I was registered as female, but I am male", "At birth I was registered as male, but I am female", and "Different from the options above, namely...". Participants who chose the final option were required to fill in a short box describing their gender identity. In total, there were 15,527 cisgender males, 22,375 cisgender females, 18 transwomen, 17 transmen and 18 individuals who chose the other option and identified with other gender identities (e.g., genderfluid). Thus, in total, there were 53 transgender and gender-diverse individuals (Supplementary Table S1).

**LifeLines: ascertaining autism diagnosis**. Autism diagnosis was ascertained using the question: "Do you have an autism diagnosis?" followed by "In what year was this diagnosed". 439 individuals indicated that they had an autism diagnosis (252 cisgender males, 184 cisgender females, and 3 transgender and gender-diverse individuals) (Supplementary Table S1).

**LifeLines: measures of traits related to autism**. All participants also completed the AQ-10[73], provided the age when they completed the AQ-10.

**Ethics**. The Human Biology Research Ethics Committee, University of Cambridge, provided ethical approval for the collection and use of data for both the APHS and the IMAGE cohorts. They also provided ethical approval to access de-identified data from the LifeLines cohort. The Psychology Research Ethics Committee of the University of Cambridge confirmed that formal ethical review was not needed for use of the C4 dataset since it was secondary use of deidentified and anonymized data. The same was confirmed for the MU dataset by the Ethical & Independent Review Services. Informed consent was obtained for all participants included in this study.

**Statistical analyses: rates of autism diagnosis**. In all five datasets, we investigated if rates of autism diagnosis significantly differed by gender by first conducting $\chi^2$ tests (Model 1, unadjusted), and then by conducting logistic regressions adjusted for age and educational attainment as covariates (Model 2, adjusted). Both age and educational attainment were associated with autism diagnosis, with younger individuals more likely to receive an autism diagnosis[78,79], and educational attainment typically negatively correlated with autism[51]. Further, these two variables were measured across all five datasets. In addition, for the IMAGE dataset, we included a dummy variable for the two studies participants were drawn from (mathematical ability and case–control) to account for potential confounding effects of recruitment.

Each model was conducted first by using three gender categories (transgender and gender-diverse, male, and female), and then by using two gender categories (transgender and gender-diverse and cisgender). Regression betas were converted to ORs. As an additional sensitivity analysis, only in the MU dataset, we repeated the analyses after dividing the cohort into four groups ("Male", "Female", "Transgender", and "Other"), to investigate if these results differed by gender identity.

Additionally, we also investigated if rates of transgender and gender-diverse individuals vary by autism diagnosis. This was done by using a logistic regression comparing transgender and gender-diverse individuals to cisgender individuals (dependent variable). Autism diagnosis was the independent variable, and educational attainment and age were included as covariates.

Whilst information for this study from all five datasets were collected using internet-based surveys, there are differences between them. Of importance is that sex, gender, and autism diagnosis information were all collected differently in the five datasets. In the C4 and MU datasets, gender information was collected using a single question whereas in the IMAGE and APHS datasets, gender information was collected using two questions—one for sex assigned at birth and another for gender identified with. In the LifeLines dataset, gender information was collected using a single question, but this included options about sex assigned at birth alongside gender. Further, information on autism diagnosis was also collected differently with deeper information provided by participants in the IMAGE, LifeLines, and APHS datasets. There are other cohort-based differences as well. For example, the MU dataset was aggregated over a long period of time and primarily collected from

the US, whilst three datasets (C4, APHS, and IMAGE) were collected over a shorter period of time and primarily from the UK. The LifeLines dataset used here was a subset of a cohort study, where participants were invited through general practitioner clinics rather than via the internet. This was collected in the Netherlands and consists of older participants.

Given the heterogeneity in these datasets, we wanted to investigate if the ORs obtained across the five datasets are comparable. Two factors affect ORs: winner's curse which inflate ORs in smaller cohorts[35,36], and lower precision, i.e., higher standard errors of ORs in smaller cohorts[80]. Thus, ORs are not directly comparable between the datasets. In order to make the ORs comparable, we generated sub-datasets of equivalent sample sizes to the three smaller datasets (IMAGE, APHS, and LifeLines) in the two larger datasets (C4 and MU). We used a subsampling bootstrap approach to compare ORs in the two larger datasets with ORs in the smaller datasets. We generated six sets of 10,000 random subsamples each from the C4 and the MU datasets. Each of the 10,000 subsamples was matched to the numbers of cisgender males, cisgender females and transgender and gender-diverse individuals in the IMAGE, APHS, and LifeLines datasets. Thus, we sampled 10,000 times from the C4 and MU datasets with each sample consisting of 766 cisgender males, 1383 cisgender females, and 162 transgender and gender-diverse individuals to match the APHS dataset. In addition, we also sampled 10,000 times from the C4 and MU datasets with each sample consisting of 994 cisgender males, 747 cisgender females, and 62 transgender and gender-diverse individuals to match the IMAGE dataset. Finally, we sampled 10,000 times from the C4 and MU datasets with each sample consisting of 15,527 cisgender males, 22,375 cisgender females, and 52 transgender and gender-diverse individuals to match the LifeLines dataset. In each sample, we calculated adjusted ORs using logistic regression. We then calculated the empirical $p$ values for the adjusted ORs for the IMAGE, APHS, and LifeLines samples from the distribution of ORs generated in the 10,000 samples from MU and C4. We corrected for the six tests using Bonferroni correction (empirical $p$ value alpha = 0.008).

**Statistical analyses: rates of other neurodevelopmental and psychiatric conditions**. In the C4 and MU datasets we investigated if diagnosis of six neurodevelopmental and psychiatric conditions differed by gender using $\chi^2$ tests (Model 1) and logistic regression accounting for educational attainment and age (Model 2). Additionally, we repeated Model 2 after excluding autistic individuals (Model 3), as there may be an autism-based ascertainment bias in these cohorts. Each model was conducted first by using three gender categories (transgender and gender-diverse, cisgender male, and cisgender female), and then two categories (transgender and gender-diverse and cisgender).

We also investigated the relative association between each neurodevelopmental and psychiatric conditions to gender identity. Gender identity (transgender and gender-diverse versus cisgender) was the dependent variable. The independent variables were diagnosis of ADHD, autism, bipolar disorder, depression, learning disorder (only in C4 dataset), OCD, and schizophrenia. Age and educational attainment were included as covariates.

**Statistical analyses: traits related to autism**. In the C4 dataset, we investigated differences in scores by gender (cisgender males, cisgender females, and transgender and gender-diverse) on the four measures using ANOVA and then conducted post-hoc $T$-tests. We repeated the analyses using linear regression accounting for age and educational attainment. Distributions in "Brain Types" between the three genders were investigated using $\chi^2$ tests. Validation using the AQ-50[77] was conducted in the IMAGE dataset, and using the AQ-10 was conducted in the LifeLines dataset.

**Statistical analyses: calculation of "Brain Types"**. Calculation of "Brain Types" was only done in the C4 dataset. We first calculated the standardized scores of the SQ-10 and the EQ-10. This was done by subtracting the mean of the SQ-10 and the EQ-10 (means were calculated using only non-autistic individuals from the C4 dataset) from each individual's score and then dividing by the maximum possible score (20 for both the SQ-10 and the EQ-10). We next calculated a "D-score" by subtracting the standardized EQ-10 score from the SQ-10 score. We then divided individuals into five Brain Types based on D-score percentiles. The lowest 2.5th percentile was Extreme Type E and the highest 2.5th percentile was Extreme Type S. Those scoring between the 35th and 65th percentiles were classified as Type B. Participants who scored between the 2.5th and 35th percentiles were Type E, and Type S was defined by scoring between the 65th and 97.5th percentile.

**Statistical analyses: multiple testing correction**. Across all the datasets and the three aims and the exploratory aim, we conducted at least 182 different analyses. Given the size of the datasets used, the standard errors are low. We thus define a study-wide $p$-value of 0.0002 to correct for all the tests. Details of the tests conducted are provided in Supplementary Table S11.

**Statistical analyses: power calculations in the LifeLines dataset**. Given the relatively low number of transgender and gender-diverse individuals, we conducted power calculations to investigate if the LifeLines cohort had sufficient statistical power to identify effects. We used effect sizes obtained from the results of the C4 dataset as this was the largest dataset, and hence, likely to have effects that are least

affected by winner's curse ("Supplementary Methods"). Power calculations suggested that we were underpowered to detect effects at an alpha of 0.05 for calculating ORs using logistic regression, with power achieved between 0.62 (reference group: cisgender males)—0.69 (reference group: cisgender females). However, we proceeded with the analyses to identify if the effects observed were in the same direction as those observed in other datasets.

**Reporting summary**. Further information on research design is available in the Nature Research Reporting Summary linked to this article.

## Data availability
As participants did not consent for their data to be publicly shared, even anonymized, data will be made available to only potential collaborators with ethical approval after they submit a research proposal to the Autism Research Centre, University of Cambridge, UK for four of the datasets (C4, MU, IMAGE, and APHS). Data for LifeLines can be obtained by making an application to the LifeLines Biobank (https://www.lifelines.nl/researcher). A reporting summary for this Article is available as a Supplementary information file.

## Code availability
Scripts are provided at: https://github.com/autism-research-centre/Atypical-gender-and-autism. All analyses were conducted using R version 3.4.4 (2018-03-15).

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

## Acknowledgements

This study was supported by the Medical Research Council (MRC), the Wellcome Trust (214322/Z/18/Z), the Templeton World Charity Foundation, the Autism Research Trust, and the National Institute of Health Research (NIHR) Collaboration for Leadership in Applied Health Research and Care-East of England (CLAHRC-EoE). The views expressed are those of the authors and not necessarily those of the NHS, the NIHR or the Department of Health. The authors also received funding from the Innovative Medicines Initiative 2 Joint Undertaking (JU) under grant agreement No 777394. The JU receives support from the European Union's Horizon 2020 research and innovation program and EFPIA and Autism Speaks, Autistica, SFARI. Funding for the Autism and Physical Health Survey was provided by the Autism Research Trust, the Rosetrees Trust, the Cambridgeshire and Peterborough NHS Foundation Trust, and the Corbin Charitable Trust. Thanks also to the Cambridge Autism Research Database, Autistica's Discover Network, and various autism support groups and charities for assisting our recruitment for the APHS. Varun Warrier is supported by the Bowring Research Fellowship at St. Catharine's College, Cambridge. D.M.G. was supported in part by the Zuckerman STEM Leadership Program. M.-C.L. is supported by the Academic Scholars Award from the Department of Psychiatry, University of Toronto, the Ontario Brain Institute via the Province of Ontario Neurodevelopmental Disorders (POND) Network (IDS-I l-02), the Canadian Institutes of Health Research (CIHR) (PJT 159578 and a CIHR Sex and Gender Science Chair, GSB-171373), and the Slaight Family Child and Youth Mental Health Innovation Fund via the CAMH Foundation. We are grateful to all the participants, and for Channel 4 for sharing the anonymized data with us.

## Author contributions

V.W. conducted the analyses. V.W. and S.B.-C. designed the study. V.W., D.M.G., E.W., C.B., P.L.S. collected the data. V.W., D.M.G., E.W., C.B., P.L.S., M.-C.-L., C.L.A., and S.B.-C. interpreted the data, wrote, read, and edited the paper.

## Competing interests

The authors declare no competing interests.
