## [Peer Review File · Nature Communications]

Reviewers' comments:

Reviewer #1 (Remarks to the Author):

Thank you for the opportunity to review and comment on the study manuscript, "Elevated autism diagnoses and autistic traits in transgender individuals: a study of over 600,000 individuals." This study makes an important contribution to the autism and gender-diversity literatures by providing compelling evidence of a link between neurodiversity and gender diversity. It does so with a convincing research design by reporting on four independent large-scale studies, each conducted with a different recruitment and assessment approach. The consistent finding of over-occurring autism among self-identified gender-diverse participants in each of the four contrasting study samples is notable. Further, this study presents the largest sample of gender-diverse people to date who have reported on co-occurring neurodiversity-related characteristics/diagnoses. The over-occurrence of autism and autism spectrum characteristics among gender-diverse individuals is a topic of interest to the autism and gender communities, and community expressions of the co-occurrence abound (e.g., the recent name change of the Autistic Women's Network to the Autistic Women and Nonbinary Network in response to the gender-diversity of their constituency). This study advances our understanding of the autism and gender-diversity co-occurrence as a phenomenon that exists in the general population, not just in gender-specialist clinical settings. This work will have an important impact by further recognizing this under-studied and poorly understood community and encouraging increased research to advance knowledge of this community's challenges and needs.

Additional noted strengths of the manuscript:

- Adjusting for educational attainment and age in analyses of ASD diagnoses
- Consideration of how odds ratios function in datasets of varying size. Use of bootstrapping procedures to render the differently-sized datasets more "comparable" for comparing ORs
- Investigation of broader psychiatric and neurodiversity characteristics in gender-diverse individuals, a topic which has been rarely studied
- Comparison of groups based on male/female/gender-diverse as well as the collapsed cisgender/gender-diverse groupings to account for different rates of autism in cisgender males and females
- Consideration of autism identity by diagnosis as well as self-report (see the Autistic Self Advocacy Network's definition of autism as a diagnosis and also an identity: <https://autisticadvocacy.org/about-asan/about-autism/>)
- Careful consideration of alternative explanations for the findings in the discussion

There are several ways in which the current manuscript could be improved:

First, it is incorrect to characterize people who are not autistic as "neurotypical". Instead, one might describe a non-autistic person as "allistic". The authors should carefully review the manuscript for this mischaracterization.

There is a concern with the use of the convenience term, "transgender" to represent people with diverse gender-identities. The authors describe a range of gender-identities and yet collapse them under the umbrella term "transgender". This is not a precise use of the term, and therefore, could be experienced as disrespectful. More accurate terms would be: gender-diverse, gender minority, or transgender/gender-diverse. It is recommended that the study team carefully consider their use of gender-related language. It is appropriate to collapse gender-diverse individuals into one category, but the term used to describe this group of people needs to be broader/more inclusive than "transgender".

Although an impressive sample across the four studies, the phrase "over 600,000 individuals" as used in the title is a bit misleading. It is true that over 600,000 people participated in the four studies, but when summed, the study sample of gender-diverse individuals (which is the group referred to in the title) is ~3,336. Perhaps the title would best focus on the inclusion of the four large study groups instead of "600,000 individuals".

The literature reviewed in the introduction is appropriate, but the characterization of existing studies as lacking misses the strength of this growing literature. The existing literature is heterogeneous -- many different settings, study approaches, nationalities, and age ranges, all showing an over-occurrence of autism and gender-diversity regardless of study approach. In fact, the current study under review has as its primary strength this same quality of heterogeneity of study samples, and it is an amplification and extension of the current available literature.

Of concern, this study is presented without any indication of the lived experiences, challenges, resilience factors, or needs of the neurodiverse gender-diverse communities. As might be imagined, autistic transgender people have struggled for recognition and rights, and transgender autistic self-advocates describe the double-diversity characteristics as a double-disparity, and times, a profound challenge. Consider, for example, Kayden Clark, the autistic transgender man in the United States who was denied gender care because of his autism diagnosis and who ended up dead after an altercation with the police related to his distress over lack of appropriate gender care. I believe there is an ethical demand when presenting data on this highly vulnerable and poorly understood community to include some references to the lived experience of being gender diverse and neurodiverse. Here are some references to learn more about this community. The study authors may wish to integrate some of these themes into their discussion:

a. This study captured the experiences and voices of gender-diverse autistic youth and was co-conducted by autistic gender-diverse research collaborators:

Strang, J.F., Powers, M.D., Knauss, M., Sibarium, E., Leibowitz, S.F., Kenworthy, L., ... Anthony, L.G. (2018). "They thought it was an obsession": Trajectories and perspectives of autistic transgender and gender-diverse adolescents. *Journal of Autism and Developmental Disorders*, 48(12), 4039-55.

b. This position statement on the rights of autistic transgender/gender-diverse people was published by the Autistic Self Advocacy Network, the National Center for Transgender Equality, and the National LGBTQ Taskforce:

https://autisticadvocacy.org/wp-content/uploads/2016/06/joint_statement_trans_autistic_GNC_people.pdf

And finally, the clinical care needs of this community should be considered in the discussion. There has been some work done with regards to the care needs of individuals with the co-occurrence, including:

a. Initial clinical care guidelines for autistic gender-diverse youth developed by an international team of experts on the co-occurrence:

Strang, J.F., Meagher, H., Kenworthy, L., de Vries, A.L.C., Menvielle, E., Leibowitz, S., ... Anthony, L.G. (2016). Initial clinical guidelines for co-occurring autism spectrum disorder and gender dysphoria or incongruence in adolescents. *Journal of Clinical Child and Adolescent Psychology*, 47(1), 105-15.

b. Recognition by the World Professional Association for Transgender Health (WPATH) that the autism

and gender-diversity co-occurrence should be part of the Global Education Initiative (GEI) training for transgender health care certification. See GEI: <https://www.wpath.org/about-gei>

Reviewer #2 (Remarks to the Author):

Summary of paper:

This paper collects data from four large and very large data sets collected in relation to questions unrelated to the present, which are

'Is autism more prevalent in the transgendered?',

'Are autistic traits elevated in the transgendered?',

'Do those who express being transgendered display elevated neurodevelopmental and psychiatric conditions?', and an exploratory aim:

'Are there higher rates of suspected undiagnosed autism in the transgendered?'

The authors have access to two extremely large internet based data sets, and two moderately sized ones, all of which were not collected to answer the present question, and three of which were collected in relation to questions surrounding autism, and so are not random in their nature, while one was not collected in relation to autism, but in relation to another question, and is therefore unlikely to be biased toward autism, but is nonetheless, not a random data set. While each of the data sets collected information pertinent toward autism, none collected information specifically to assess for transgenderism, instead inferring this on the basis of responses to questions about sex. The results find that in all four data sets, there are:

Elevated odds ratios for autism in the assigned transgendered individuals.

Transgendered report scores for instruments suggesting elevated autistic traits

Transgendered individuals report increased diagnoses for other psychiatric and neurodevelopmental conditions

The Transgendered group reported a higher rate of suspected diagnosis than males, females, and cisgendered as a whole.

It is concluded that transgendered individuals are more likely to be autistic than the cisgendered.

What this study has not revealed is whether there is a higher rate of transgenderism within autism, and surely the data could have been interrogated for this.

Line 73 to 74 a criticism is made of previous studies of adults that having samples less than 500 may have biased the odds ratio. While this is correct. 1/. A reference is needed for this claim. 2/. the bias is proportional to $1/n$, so that in larger samples the bias will be smaller (Nemes et al., 2009; Zhong & Prentice 2008, 2010). This should be recognised, and that in no sample is the bias = 0.

Lines 167 to 171. While it might be argued that individuals would respond with their gender, and not their sex, those who are gender aware may have a higher propensity to respond with their actual sex. This may be their attained sex, or their sex assigned at birth. Hence, some transgendered may have appeared as cisgendered in this sample, biasing the results. If it is correct that the actual

transgendered have a heightened rate of autism, then this assumption will underestimate the presence of autism in this group. If it is false, and something in those identifying as transgendered (such as reduced social awareness) increased their propensity to so self-identify, then these results will over-estimate the presence of autism in this group. Whichever it is, is unclear, and cannot be clarified by this approach. It is possible this analysis has under or over identified autism among the transgendered. However, the sheer magnitude of the samples suggests some robustness.

Line 187. It is suggested that the wording of the questions around diagnosis might preclude individuals who have not received a formal diagnosis from claiming a self-diagnosed condition. However, It may, but this may also be biased by diagnosis, with some diagnoses being associated with differing levels of self-diagnosis and claiming this.

Lines 214 to 224. The discussion indicates that other gender was combined with transgender for the MU data set. It is argued that some transgendered individuals may identify as other, and so have chosen other. If so, why was this not undertaken for the C4 dataset (cp: lines 177 to 180)?

Lines 228 to 229. Why were only 54,127 individuals asked about formal diagnoses and the other 31,543 individuals in the MU data set were not?

Line 323 to 324: A reference is needed for this claim.

Lines 357 to 369 The term "Brain type" will not be clear to those readers unfamiliar with empathizing and systematizing theory associated with the EMB hypothesis. Either place some introductory material toward this, or use language less likely to confuse, such as cognition type.

Line 379 to 380 This is not presently a functioning link (22nd September) <https://github.com/autism-research-centre/Atypical380-gender-and-autism>

In several places such as Line 392 or 404 sentences are commenced "X2 test" (the absence of χ^2 maybe a typeface issue though, and so use a typeface that reveals the Greek letter). First please correct this ungrammatical statement throughout to "A Chi-square test" or "A χ^2 test", using a definite article and the symbol for chi. (Sorry for being pedantic.)

Throughout the text p-values are reported with scientific notation. For instance, line 393. It is not appropriate when you set a criterion p-value (Line 375) to report actual p-values, as these may become confused with effect sizes, which they are not, as p-values are strongly influenced by sample size. This occurs throughout the text, and would be better addressed by reporting actual relevant effect sizes. In the present instance, phi would be the effect size, and is equal to 0.103. Further, the chi-value looks very impressive, however, Chi is always proportional to the sample size used, and as there were 514,100 participants in this sample, a chi = 3316 is actually not overly impressive.

Line 466 to 468 Wouldn't a MANOVA be the more appropriate test of this hypothesis here, followed by the univariate analyses. As these are evaluations of gender differences, d-values need to be reported as estimates of effect size, not simply stating the extreme p-value.

Figure 3 A brief explanation as to why the density plots display waves at integers for the NT male and NT female data, but not the NT transgender data would be useful

On line 571, us a definite article preceding "Chi-square"

Discussion

I would like to see a clear summary of the results and a definite statement of how they bear upon the hypotheses. At present, there is a statement that there is an elevated risk of being autistic for the transgendered.

Line 658 to 661 the suggestion that "life experiences of being transgender, not fitting in with stereotypical social and gender norms, may contribute to elevated rates of certain related conditions such as depression in transgender individuals" has been, in part, addressed by a paper by George and Stokes other than the one already cited.

Lines 665 to 669 Claim: "Second, there is a possibility that some nonbinary, gender-neutral, or other gender diverse individuals may not identify with the 'transgender' term in the C4 dataset as we did not concurrently provide the 'transgender' and 'other' options. However, this differential misclassification is likely to attenuate the effect size between transgender identity and autism by increasing the rate of autism in the cisgender category"

There is also a possibility that some individuals who have transitioned from male to female or female to male will identify as their target gender, attenuating your results. It is possible that those identifying as transgendered are more likely to be socially non-conforming, and possibly manifesting other symptoms. By not identifying the gender assigned at birth and finding if this varies from the gender reported presently, you cannot exclude this, and that, as such, a proportion of your actually transgendered sample have not been captured and this may have amplified your results.

Also, I don't think you can conclude here that because some diverse individuals having not identified with the term 'transgender' would amplify your results had they been included in the transgender group. Such individuals may have also been more or less likely to be autistic, or no different. I suggest you temper this conclusion accordingly.

Lines 673 to 675 "Further, as pointed out earlier, the ORs are similar between the four internet-based datasets in this study and a study based on GD-clinic based samples". How does this differ from what is presented in lines 670-671 "Further, subsampling bootstrap analyses 670 indicate that the ORs are similar across the different datasets"? You could just as easily refer to study 7's GD-clinic based sample there.

Lines 675 to 676 "Fourth, individuals with severe mental health conditions and intellectual disability are less likely to participate." What is the basis for this claim? Additionally, even where it true, which I see no reason to accept on the basis of mere assertion, what is the importance of this? How would those with ID or mental health concerns not participating influence the results?

References:

George, R. & Stokes, M.A. (2018). A Quantitative Analysis of Mental Health among Sexual and Gender Minorities in ASD. *Journal of Autism & Developmental Disorders*, 48(6): 2052-2063.
<http://doi.org/10.1007/s10803-018-3469-1>

Nemes, S., Jonasson, J.M., Genell, A. & Steineck, G. (2009). Bias in odds ratios by logistic regression modelling and sample size. *BMC Medical Research Methodology*, 9, 56-60. doi: 10.1186/1471-2288-9-56.

Zhong, H. & Prentice, R.L. (2008). Bias-reduced estimators and confidence intervals for odds ratios in genome-wide association studies. *Biostatistics*, 9(4), 621-34. doi: 10.1093/biostatistics/kxn001

Zhong, H. & Prentice, R.L. (2010). Correcting "winner's curse" in odds ratios from genomewide association findings for major complex human diseases. *Genetic Epidemiology*, 34(1), 78-91. doi:10.1002/gepi.20437.

Mark A Stokes

Reviewer #3 (Remarks to the Author):

This study makes use of several different data sets to test the hypothesis that transgender status is associated with autism. Other mental disorders are also considered.

I have some relatively minor concerns and one major concern, all of which I expect can be addressed.

Autism diagnoses are mentioned in the title, but are not mentioned in the abstract and should because clearly this was part of the study.

The introduction is helpful in regard to autism although (see below) hardly touches on how other mental disorders may play a role.

Methods

The survey sources could also be cited in the introductory text (although they are clearly cited in the methods).

In the IMAGE data set a case-control design was used different from the other survey (cohort) based designs. Estimates could be altered by the design and if combined should be a taken into account in any models. Was this addressed and if so how?

Analysis

The heading 'rates of autism' seems misleading - my assumption is this should be 'rates of autism diagnoses' as described earlier. This term should be used consistently throughout as the media and non technical readers may pick this up incorrectly assuming it to mean autism rates (in adulthood autism population rates and rates of autism diagnosis are likely to be very different). Autism diagnoses were available from some of the studies only. Traits were obtained from the remainder (the subsequent subheading). The term autism diagnoses is later used in the same paragraph. This lack of precise terminology is unusual in this otherwise quite precisely drafted paper.

Lack of comparability is mentioned further down and how it was handled using bootstrap sampling - given the heterogeneity of the data sources more (or clearer) information on how such differences were addressed would be helpful.

Brain types is not explained in the methods (or indeed the introductions), which may be a rather specialist term of interest to the authors. And why is it relevant or necessary? Unless clearly justified it could be dropped?

Results

Findings in Figure 4a are not described in the text. There is not a description of associations of gender identity with autism (diagnoses/traits) compared with the other conditions studied. To the non specialist eye, except for schizophrenia, there is no difference between disorder associations in figure 4. In figure 4a autism seems to stand out (we are not told is this traits or diagnoses).

Key concern.

If the association of gender is not specific to autism the article title should reflect this. And it would no longer be a surprising or original finding as transgender people are more likely to be distressed or mentally unwell (according to studies of people presenting for medical or surgical interventions). All of the neuropsychiatric conditions studied include mental distress (even intellectual disability). It is unsafe to presume that each is separately and accurately delineated using the methods described; a common distress dimension may possibly explain the underlying thrust of the study findings. Thus, a devil's advocate argument is that all this paper shows is that mental distress (of various kinds) is associated with declared transgender status. This would be old news, albeit on a more substantial scale (adding something to the literature but not a major finding). The paper does not convincingly show that the association with gender is specific to autism. It suggests therefore an already known non specific association of mental morbidity with transgender status (both self described). There may be opportunities here to clarify this issue or conduct further analyses addressing this issue. The study is unusual (and particularly good) in examining associations with a range of conditions in addition to autism (and not just autism alone).

The authors seem to touch on this issue but not tackle it head on in the discussion in the paragraph: "Our study also identified that transgender individuals have elevated rates of autism-related neurodevelopmental and psychiatric conditions in two datasets when compared to cisgender individuals". Interestingly they state that "In contrast to the ORs for autism, the ORs for these related conditions varied between the two datasets. While the OR for schizophrenia in transgender individuals was high in the C4 dataset, it was low and failed to reach statistical significance in the MU dataset." This argument touches on but does not tackle sufficiently my main concern, which could presumably be formally modelled. Their position on this question then seems to emerge in the final concluding statement and in a discussion comment on causality where they say that 'a few hypotheses may explain the over-representation of autism and related conditions in transgender individuals...' [note 'and']. If that is actually what this study shows that should be stated in the title (which as I said earlier seems to suggest that the over-representation is specific to autism). The discussion in this paragraph includes quite interesting causal explanations overlooking the more parsimonious explanation that underlying distress (however that may be described or characterised), explains the main finding, common to all these conditions.

An omission is any reference to the literature from transgender health clinics and intervention studies showing elevated levels of mental distress and disorder in such populations, i.e. arguably this is already known although this is a more thorough and substantial study of this question.

Apart from this the paper describes possible biases and limitations in a balanced and transparent way. I would be very interested in seeing a revision addressing the concerns I have raised.

Reviewer:

Traolach Brugha, University of Leicester, 4 October, 2019.

Reviewer #4 (Remarks to the Author):

Thank you for the opportunity to review this paper. The topic of gender variation among individuals with autism is important to understand so that providers can better identify areas of benefit and need. Unfortunately there are several limitations of the current study that prevent me from recommending publication at this time. I provided detailed feedback on the introduction and method section to help

guide future revisions of the paper.

Introduction

The introduction needs revision to provide focus and clear definition of terms. It would be helpful to start the paper with a brief discussion of gender identity as a continuous construct and the range of possible identities commonly identified in the literature. This overview would help the reader distinguish between the broader idea of gender variation (i.e., gender identity that does not always correspond to sex assigned at birth) from specific types of variations (e.g., transgender, non-binary, gender fluid, and gender queer). Along these same lines, it is not appropriate to include terms such as non-binary, gender fluid, and gender queer as examples of transgender identity since these terms represent individuals at different points on the sex/identity spectrum.

The introduction would be strengthened if authors provided estimates of (1) the number of gender variant and transgender people in the general population and (2) the number of gender variant and transgender people among those with autism; followed by

- (1) The number of individuals with autism in the general population and
- (2) the number of people with autism in the gender variant and transgender population.

Following, limitations of previous studies could be introduced, including small clinic-based samples, ascertainment of gender variation from one checklist item on the CBCL, and defining gender variation by endorsement of gender dysphoria.

The details, strengths, and limitations of the datasets used in these analyses belong elsewhere in the paper (i.e., method and discussion).

Method

Only one of the four datasets provided can answer the research questions with acceptable methods. The other three datasets have significant limitations that prevent interpretation of results. As mentioned previously, gender identity (e.g., cisgender, transgender, non-binary) is distinctly different from biological sex assigned at birth (i.e., male, female, intersex). In the Channel 4 and Music studies, participants were asked about sex but not about gender. Moreover, when asked about sex, participants were given two options that could be considered both sex and gender (i.e., male and female) and one option that could be considered only gender (transgender). In the physical health survey, participants were asked about sex and gender but the response options for gender were male, female, non-binary, and other. Therefore, the physical health survey data can only be used to estimate gender variance and/or non-binary status.

The genetics dataset is the only one included in the study that asked about sex and gender and had appropriate response options for both stem questions. I would therefore include only the genetics dataset in future analyses. It is my opinion that the cost of limiting the sample to one dataset is superseded by the benefits of sound methodology.

Reviewers' comments:

Reviewer #1 (Remarks to the Author)

Thank you for the opportunity to review and comment on the study manuscript, “Elevated autism diagnoses and autistic traits in transgender individuals: a study of over 600,000 individuals.” This study makes an important contribution to the autism and gender-diversity literatures by providing compelling evidence of a link between neurodiversity and gender diversity. It does so with a convincing research design by reporting on four independent large-scale studies, each conducted with a different recruitment and assessment approach. The consistent finding of over-occurring autism among self-identified gender-diverse participants in each of the four contrasting study samples is notable. Further, this study presents the largest sample of gender-diverse people to date who have reported on co-occurring neurodiversity-related characteristics/diagnoses. The over-occurrence of autism and autism spectrum characteristics among gender-diverse individuals is a topic of interest to the autism and gender communities, and community expressions of the co-occurrence abound (e.g., the recent name change of the Autistic Women’s Network to the Autistic Women and Nonbinary Network in response to the gender-diversity of their constituency). This study advances our understanding of the autism and gender-diversity co-occurrence as a phenomenon that exists in the general population, not just in gender-specialist clinical settings. This work will have an important impact by further recognizing this under-studied and poorly understood community and encouraging increased research to advance knowledge of this community’s challenges and needs.

Additional noted strengths of the manuscript:

- Adjusting for educational attainment and age in analyses of ASD diagnoses
- Consideration of how odds ratios function in datasets of varying size. Use of bootstrapping procedures to render the differently-sized datasets more “comparable” for comparing ORs
- Investigation of broader psychiatric and neurodiversity characteristics in gender-diverse individuals, a topic which has been rarely studied
- Comparison of groups based on male/female/gender-diverse as well as the collapsed cisgender/gender-diverse groupings to account for different rates of autism in cisgender males and females
- Consideration of autism identity by diagnosis as well as self-report (see the Autistic Self Advocacy Network’s definition of autism as a diagnosis and also an identity: <https://autisticadvocacy.org/about-asan/about-autism/>)
- Careful consideration of alternative explanations for the findings in the discussion

We thank you for your careful and positive appraisal of the manuscript. We have endeavoured to address all your comments below.

There are several ways in which the current manuscript could be improved:

1. First, it is incorrect to characterize people who are not autistic as “neurotypical”. Instead, one might describe a non-autistic person as “allistic”. The authors should carefully review the manuscript for this mischaracterization.

We agree that the term “neurotypical” can be misleading here - for instance non-autistic individuals with ADHD are not ‘neurotypical’ but allistic. However, we are a bit concerned that the term ‘allistic’ will not be familiar to many readers outside of the autism field. We have thus decided to use the term “non-autistic” throughout the manuscript instead of “neurotypical”.

2. There is a concern with the use of the convenience term, “transgender” to represent people with diverse gender-identities. The authors describe a range of gender-identities and yet collapse them under the umbrella term “transgender”. This is not a precise use of the term, and therefore, could be experienced as disrespectful. More accurate terms would be: gender-diverse, gender minority, or transgender/gender-diverse. It is recommended that the study team carefully consider their use of gender-related language. It is appropriate to collapse gender-diverse individuals into one category, but the term used to describe this group of people needs to be broader/more inclusive than “transgender”.

Thank you for the very helpful reminder and suggestion. We agree that the term “transgender” does not encompass the range of gender identities we have currently included under the umbrella definition “transgender”. We have followed the reviewer’s suggestion to use “transgender and gender diverse” to refer to the transgender and gender-diverse individuals characterized in the present study.

3. Although an impressive sample across the four studies, the phrase “over 600,000 individuals” as used in the title is a bit misleading. It is true that over 600,000 people participated in the four studies, but when summed, the study sample of gender-diverse individuals (which is the group referred to in the title) is ~3,336. Perhaps the title would best focus on the inclusion of the four large study groups instead of “600,000 individuals”.

We have now changed it to “Elevated rates of autism diagnoses, autistic traits, and other diagnoses in transgender and gender-diverse individuals: a study of five datasets”.

4. The literature reviewed in the introduction is appropriate, but the characterization of existing studies as lacking misses the strength of this growing literature. The existing literature is heterogeneous -- many different settings, study approaches, nationalities, and age ranges, all showing an over-occurrence of autism and gender-diversity regardless of study approach. In fact, the current study under review has as its primary strength this same quality of heterogeneity of study samples, and it is an amplification and extension of the current available literature.

We completely agree. Thank you for pointing this out. We agree that there is a wide variety of approaches that have been used to investigate this. We have rewritten the introduction,

especially the fourth paragraph to address this reviewer concern. We hope this provides a more balanced characterization of the current literature.

Of concern, this study is presented without any indication of the lived experiences, challenges, resilience factors, or needs of the neurodiverse gender-diverse communities. As might be imagined, autistic transgender people have struggled for recognition and rights, and transgender autistic self-advocates describe the double-diversity characteristics as a double-disparity, and times, a profound challenge. Consider, for example, Kayden Clark, the autistic transgender man in the United States who was denied gender care because of his autism diagnosis and who ended up dead after an altercation with the police related to his distress over lack of appropriate gender care. I believe there is an ethical demand when presenting data on this highly vulnerable and poorly understood community to include some references to the lived experience of being gender diverse and neurodiverse. Here are some references to learn more about this community. The study authors may wish to integrate some of these themes into their discussion:

a. This study captured the experiences and voices of gender-diverse autistic youth and was co-conducted by autistic gender-diverse research collaborators:

Strang, J.F., Powers, M.D., Knauss, M., Sibarium, E., Leibowitz, S.F., Kenworthy, L., ... Anthony, L.G. (2018). "They thought it was an obsession": Trajectories and perspectives of autistic transgender and gender-diverse adolescents. *Journal of Autism and Developmental Disorders*, 48(12), 4039-55.

b. This position statement on the rights of autistic transgender/gender-diverse people was published by the Autistic Self Advocacy Network, the National Center for Transgender Equality, and the National LGBTQ Taskforce:

https://autisticadvocacy.org/wp-content/uploads/2016/06/joint_statement_trans_autistic_GNC_people.pdf

And finally, the clinical care needs of this community should be considered in the discussion. There has been some work done with regards to the care needs of individuals with the co-occurrence, including:

a. Initial clinical care guidelines for autistic gender-diverse youth developed by an international team of experts on the co-occurrence:

Strang, J.F., Meagher, H., Kenworthy, L., de Vries, A.L.C., Menvielle, E., Leibowitz, S., ... Anthony, L.G. (2016). Initial clinical guidelines for co-occurring autism spectrum disorder and gender dysphoria or incongruence in adolescents. *Journal of Clinical Child and Adolescent Psychology*, 47(1), 105-15.

b. Recognition by the World Professional Association for Transgender Health (WPATH) that the autism and gender-diversity co-occurrence should be part of the Global Education

Initiative (GEI) training for transgender health care certification. See GEI: <https://www.wpath.org/about-gei>

Thank you very much for both the points above. We completely agree that there needs to be a nuanced, ethical and sensitive discussion of the results integrating both the lived experience of autistic individuals who identify as transgender/gender diverse and incorporating the clinical needs of these individuals. We have substantially amended our discussion to incorporate both these issues, also based on the recommended references.

“These findings must be interpreted keeping in mind the lived experiences and clinical and non-clinical needs of transgender and gender diverse individuals. Both autistic individuals and transgender and gender diverse individuals represent marginalized groups where the currently available support and understanding is inadequate. Self-harm, suicidal ideation and suicide, and other vulnerabilities are high in both groups ⁴³⁻⁴⁵. This intersection of autism and gender diversity can be doubly distressing if adequate safe-guarding and support aren’t provided. A recent study demonstrated that a third of autistic individuals had their gender identity questioned because they were autistic. This co-occurrence may require specialist clinical care for autistic transgender and gender diverse individuals. Additionally, recent studies demonstrate that autistic traits differ between males and females ^{46,47}. However, it is unclear if autistic traits differ in transgender and gender-diverse individuals compared to cisgender individuals.”

Reviewer #2 (Remarks to the Author):

Summary of paper:

This paper collects data from four large and very large data sets collected in relation to questions unrelated to the present, which are

'Is autism more prevalent in the transgendered?',

'Are autistic traits elevated in the transgendered?',

'Do those who express being transgendered display elevated neurodevelopmental and psychiatric conditions?', and an exploratory aim:

'Are there higher rates of suspected undiagnosed autism in the transgendered?'.

The authors have access to two extremely large internet based data sets, and two moderately sized ones, all of which were not collected to answer the present question, and three of which were collected in relation to questions surrounding autism, and so are not random in their nature, while one was not collected in relation to autism, but in relation to another question, and is therefore unlikely to be biased toward autism, but is nonetheless, not a random data set. While each of the data sets collected information pertinent toward autism, none collected information specifically to assess for transgenderism, instead inferring this on the basis of responses to questions about sex. The results find that in all four data sets, there are:

Elevated odds ratios for autism in the assigned transgendered individuals.

Transgendered report scores for instruments suggesting elevated autistic traits

Transgendered individuals report increased diagnoses for other psychiatric and neurodevelopmental conditions

The Transgendered group reported a higher rate of suspected diagnosis than males, females, and cisgendered as a whole.

It is concluded that transgendered individuals are more likely to be autistic than the cisgendered.

1. What this study has not revealed is whether there is a higher rate of transgenderism within autism, and surely the data could have been interrogated for this.

Thank you for the review of the paper. The first version of our manuscript has provided unadjusted ORs for this question. Now, we provide adjusted ORs for the rates of transgender and gender diverse individuals within autism, after adjusting for age and educational attainment.

2. Line 73 to 74 a criticism is made of previous studies of adults that having samples less than 500 may have biased the odds ratio. While this is correct. 1/. A reference is needed for this claim. 2/. the bias is proportional to $1/n$, so that in larger samples the bias will be smaller (Nemes et al., 2009; Zhong & Prentice 2008, 2010). This should be recognised, and that in no sample is the bias = 0.

We agree with the reviewer and have now provided the references to support our argument. We have also suggested that no sample will have a bias = 0 unless the sample is the population.

3. Lines 167 to 171. While it might be argued that individuals would respond with their gender, and not their sex, those who are gender aware may have a higher propensity to respond with their actual sex. This may be their attained sex, or their sex assigned at birth. Hence, some transgendered may have appeared as cisgendered in this sample, biasing the results. If it is correct that the actual transgendered have a heightened rate of autism, then this assumption will underestimate the presence of autism in this group. If it is false, and something in those identifying as transgendered (such as reduced social awareness) increased their propensity to so self-identify, then these results will overestimate the presence of autism in this group. Whichever it is, is unclear, and cannot be clarified by this approach. It is possible this analysis has under or over identified autism among the transgendered. However, the sheer magnitude of the samples suggests some robustness.

We agree with the reviewer's view here and have amended the discussion to reflect this: "Second, there is a possibility that some nonbinary, gender-neutral, or other gender diverse individuals may not identify with the 'transgender' term in the C4 dataset as we did not concurrently provide the 'transgender' and 'other' options. Further, some gender-aware individuals may respond by providing their assigned sex rather than their gender. It is difficult to disentangle this. However, the magnitude of the sample size suggests that the effects of such misclassification will have a minimal effect on the analyses. Further, subsampling bootstrap analyses indicate that the ORs are similar across the different datasets."

4. Line 187. It is suggested that the wording of the questions around diagnosis might preclude individuals who have not received a formal diagnosis from claiming a self-diagnosed condition. However, it may, but this may also be biased by diagnosis, with some diagnoses being associated with differing levels of self-diagnosis and claiming this.

We agree. We have now inserted "though not completely eliminate" in parenthesis. Thus, we are not suggesting that there won't be any self-diagnosed autistic individuals in the study, but rather, this will be minimized based on how the question was worded.

5. Lines 214 to 224. The discussion indicates that other gender was combined with transgender for the MU data set. It is argued that some transgendered individuals may identify as other, and so have chosen other. If so, why was this not undertaken for the C4 dataset (cp: lines 177 to 180)?

The difference between the MU dataset and the C4 dataset is that in the MU dataset, the "Transgender" and the "Other" option were provided concurrently. In contrast, in the C4 dataset, these were provided sequentially. Initially, the four options used whilst collecting data for the C4 dataset were "Male", "Female", "Transgender" and "Prefer not to say". This was later modified to "Male", "Female", "Other" and "Prefer not to say" to make it more inclusive. As this was done sequentially, and as "Other" and "Transgender" may have different connotations to different people, in the C4 dataset we restricted the analyses only to individuals who were provided with "Transgender" as one of the four options.

6. Lines 228 to 229. Why were only 54,127 individuals asked about formal diagnoses and the other 31,543 individuals in the MU data set were not?

This was due to how the questionnaire was developed. The question of autism diagnosis was asked a few questions above the question pertaining to diagnosis of other conditions. None of the questions were compulsory. There was significant attrition at various points in the questionnaire and the data was collected over a few years.

7. Line 323 to 324: A reference is needed for this claim.

We have now provided a reference for this.

8. Lines 357 to 369 The term "Brain type" will not be clear to those readers unfamiliar with empathizing and systematizing theory associated with the EMB hypothesis. Either place some introductory material toward this, or use language less likely to confuse, such as cognition type.

We agree and apologize for not clearly introducing this. We have now added a few lines and references introducing the "Brain types" at the first occurrence of the term in the text

"Brain types refers to an individual's cognitive profile based on their scores on their empathy and systemizing. Individuals may be classified into one of five different brain types based on their differences between their systemizing and empathy scores^{29,32}."

9. Line 379 to 380 This is not presently a functioning link (22nd September)
<https://github.com/autism-research-centre/Atypical380-gender-and-autism>

We think this is due to the line number being inadvertently included in the link (the link has been broken over two lines). The current link is <https://github.com/autism-research-centre/Atypical-gender-and-autism>

Clicking on this link leads to the github page.

10. In several places such as Line 392 or 404 sentences are commenced "X2 test" (the absence of χ^2 maybe a typeface issue though, and so use a typeface that reveals the Greek letter). First please correct this ungrammatical statement throughout to "A Chi-square test" or "A χ^2 test", using a definite article and the symbol for chi. (Sorry for being pedantic.)

Thank you for pointing this out. For the two lines, we have added the article "A". With regard to X2 and χ^2 confusion, this seems to be a type-face issue. We have not corrected the type-face issue as this depends on the typeface used at publication. We will ensure that χ^2 is well rendered upon publication.

11. Throughout the text p-values are reported with scientific notation. For instance, line 393. It is not appropriate when you set a criterion p-value (Line 375) to report actual p-

values, as these may become confused with effect sizes, which they are not, as p-values are strongly influenced by sample size. This occurs throughout the text, and would be better addressed by reporting actual relevant effect sizes. In the present instance, phi would be the effect size, and is equal to 0.103. Further, the chi-value looks very impressive, however, Chi is always proportional to the sample size used, and as there were 514,100 participants in this sample, a chi = 3316 is actually not overly impressive.

Thank you. We agree that it's important to provide effect sizes. We have provided effect sizes (and 95% CIs) for almost all tests conducted, such as Odds Ratios, regression betas or Cohen's D. We have, however, missed providing effect sizes for Chi-square tests. We have corrected this and provided effect sizes for all Chi-square tests now. We have not changed the way we have reported our p-value as it's now clearer what the effect size is for all comparisons.

12. Line 466 to 468 Wouldn't a MANOVA be the more appropriate test of this hypothesis here, followed by the univariate analyses. As these are evaluations of gender differences, d-values need to be reported as estimates of effect size, not simply stating the extreme p-value.

We are unsure if MANOVA is the appropriate test here. An ANCOVA would be appropriate if we wanted to include autism diagnosis as a covariate. But we have done a regression (which is equivalent to an ANCOVA) just after the initial univariate analyses.

We have also reported the Cohen's Ds in Supplementary Table 6 for all comparisons.

13. Figure 3 A brief explanation as to why the density plots display waves at integers for the NT male and NT female data, but not the NT transgender data would be useful.

This is almost entirely due to the sample size and smoothing whilst creating the plots in R. We used the standard smoothing parameters provided in ggplot (our code can be found in the github page). As the sample size for the neurotypical (now, non-autistic) transgender individuals is low, the smoothing behaves differently, and the density curves are more smooth as the resolution in the kernel density estimates are low. We have clarified this in the figure legend. "The non-autistic transgender kernel density plots appear more smooth due to the relatively low number of participants included, hence providing less resolution in the kernel density estimates when compared to the non-autistic males and non-autistic kernel density plots."

14. On line 571, us a definite article preceding "Chi-square"

Thank you. We have corrected this, and changed it to χ^2 in line with the rest of the text.

Discussion

15. I would like to see a clear summary of the results and a definite statement of how they bear upon the hypotheses. At present, there is a statement that there is an elevated risk of being autistic for the transgendered.

We have modified the first paragraph of the discussion to provide a clear summary of the results.

16. Line 658 to 661 the suggestion that “life experiences of being transgender, not fitting in with stereotypical social and gender norms, may contribute to elevated rates of certain related conditions such as depression in transgender individuals” has been, in part, addressed by a paper by George and Stokes other than the one already cited.

Thank you. We have now cited the paper.

17. Lines 665 to 669 Claim: “Second, there is a possibility that some nonbinary, gender-neutral, or other gender diverse individuals may not identify with the ‘transgender’ term in the C4 dataset as we did not concurrently provide the ‘transgender’ and ‘other’ options. However, this differential misclassification is likely to attenuate the effect size between transgender identity and autism by increasing the rate of autism in the cisgender category”

There is also a possibility that some individuals who have transitioned from male to female or female to male will identify as their target gender, attenuating your results. It is possible that those identifying as transgendered are more likely to be socially non-conforming, and possibly manifesting other symptoms. By not identifying the gender assigned at birth and finding if this varies from the gender reported presently, you cannot exclude this, and that, as such, a proportion of your actually transgendered sample have not been captured and this may have amplified your results.

Also, I don't think you can conclude here that because some diverse individuals having not identified with the term 'transgender' would amplify your results had they been included in the transgender group. Such individuals may have also been more or less likely to be autistic, or no different. I suggest you temper this conclusion accordingly.

Thank you, and we agree. We have amended the discussion. “Second, there is a possibility that some nonbinary, gender-neutral, or other gender diverse individuals may not identify with the ‘transgender’ term in the C4 dataset as we did not concurrently provide the ‘transgender’ and ‘other’ options. Further, some gender-aware individuals may respond by providing their sex rather than their gender. It is difficult to disentangle this. However, the magnitude of the sample size suggests that the effects of such misclassification will have a minimal effect on the analyses. Further, subsampling bootstrap analyses indicate that the ORs are similar across the different datasets.”

18. Lines 673 to 675 “Further, as pointed out earlier, the ORs are similar between the four internet-based datasets in this study and a study based on GD-clinic based samples”. How does this differ from what is presented in lines 670-671 “Further, subsampling bootstrap analyses 670 indicate that the ORs are similar across the different datasets”? You could just as easily refer to study 7's GD-clinic based sample there.

The point that we were trying to make here was that within the study, the ORs are similar based on subsample bootstrapping. This suggests that regardless of how autism or gender

identity was ascertained in the four datasets, there is a convergence of results. The second point that we were making was that while the study dependent entirely on internet-based surveys, these results are similar to those identified in GD based clinics. We have rewritten this in the discussion to make it clearer.

“Further, subsampling bootstrap analyses indicate that the ORs are similar across the different datasets. Additionally, the ORs are similar between the four internet-based datasets in this study and a study based on GD-clinic based samples¹. This similarity in results between our study and that identified in the GD-clinic based sample suggests that regardless of recruitment (internet-based vs clinic-based) or ascertainment criteria (self-report gender identity vs clinically ascertained gender-dysphoria) or age (adults vs children) the results converge on similar ORs.”

19. Lines 675 to 676 “Fourth, individuals with severe mental health conditions and intellectual disability are less likely to participate.” What is the basis for this claim? Additionally, even where it true, which I see no reason to accept on the basis of mere assertion, what is the importance of this? How would those with ID or mental health concerns not participating influence the results?

There is genetic and epidemiological evidence to suggest that individuals with mental health conditions are less likely to participate in studies or agree to be recalled (we have provided references for this now). Our datasets will have under-sampled individuals with co-occurring ID as our consent was not written for caregivers. Further, the questionnaires used in the study, especially the AQ-10, have not been developed for autistic individuals with co-occurring ID. We are not entirely sure how this will influence the results – it may not. However, we would like to state this fact in the discussion for two reasons. First, we don’t want readers to be under the impression that the results of this study, based on the methods, is applicable to all individuals on the autism spectrum. We think it is generalizable to most individuals on the spectrum, but the results may be different for autistic individuals with co-occurring ID. Second, there is increasing awareness that autistic individuals with co-occurring ID are excluded from some studies. Rather than skirting the issue, we wanted to acknowledge that autistic individuals with co-occurring ID may be underrepresented in this study and mention it as a limitation.

References:

- George, R. & Stokes, M.A. (2018). A Quantitative Analysis of Mental Health among Sexual and Gender Minorities in ASD. *Journal of Autism & Developmental Disorders*, 48(6): 2052-2063. <http://doi.org/10.1007/s10803-018-3469-1>
- Nemes, S., Jonasson, J.M., Genell, A. & Steineck, G. (2009). Bias in odds ratios by logistic regression modelling and sample size. *BMC Medical Research Methodology*, 9, 56-60. doi: 10.1186/1471-2288-9-56.
- Zhong, H. & Prentice, R.L. (2008). Bias-reduced estimators and confidence intervals for odds ratios in genome-wide association studies. *Biostatistics*, 9(4), 621-34. doi: 10.1093/biostatistics/kxn001

Zhong, H. & Prentice, R.L. (2010). Correcting “winner's curse” in odds ratios from genomewide association findings for major complex human diseases. *Genetic Epidemiology*, 34(1), 78–91. doi:10.1002/gepi.20437.

Mark A Stokes

Reviewer #3 (Remarks to the Author):

This study makes use of several different data sets to test the hypothesis that transgender status is associated with autism. Other mental disorders are also considered.

I have some relatively minor concerns and one major concern, all of which I expect can be addressed.

Thank you for reviewing our manuscript and the positive appraisal of the study.

1. Autism diagnoses are mentioned in the title, but are not mentioned in the abstract and should because clearly this was part of the study.

We have now mentioned this in the abstract.

2. The introduction is helpful in regard to autism although (see below) hardly touches on how other mental disorders may play a role.

We agree with the reviewer's reminder and have amended our introduction accordingly.

“In parallel, studies have also investigated the rates of mental health conditions and mental distress in transgender and gender diverse individuals, including individuals with GD (e.g. refs²⁻⁹). The literature has been heterogeneous with varying research methodologies and sample sizes¹⁰. Two recent literature reviews have identified higher rates of mental health conditions and mental distress (notably depression, anxiety, and substance use disorders) in transgender and gender diverse individuals compared to cisgender individuals^{5,10}. Most of the research on mental health has focused on depression, substance misuse, and anxiety, with still limited research on other neurodevelopmental and psychiatric conditions. It is also unclear currently how the elevated rates of autism diagnosis in transgender and gender diverse individuals compare to other neurodevelopmental and psychiatric conditions. To our knowledge, barring one study¹¹, none of these studies have compared the rates of other related neurodevelopmental and psychiatric conditions in transgender and gender diverse individuals compared to cisgender individuals, making it difficult to estimate if the observed effects are specific to autism.”

Methods

3. The survey sources could also be cited in the introductory text (although they are clearly cited in the methods).

Thank you. Given that the introduction is already quite long now, and that these survey sources are not well-known (and hence would require detailed explanation), we have decided that it would be best to be kept in the methods.

4. In the IMAGE data set a case-control design was used different from the other survey (cohort) based designs. Estimates could be altered by the design and if combined should be taken into account in any models. Was this addressed and if so how?

Three of our cohorts used a case-control design to test various hypotheses. These are: C4 , IMAGE, and APHS. The MU dataset consists of data collected over a longer period of time for many different online studies and tests primarily are pertaining to musical aptitude and personality. Within the IMAGE dataset, there were two nested datasets: an autism case-control dataset, and a mathematical aptitude dataset. In the latter, we invited participants who had completed or were pursuing a degree related to mathematics. This forms a part of a larger study to investigate the similarities and differences between autism, mathematical ability and variation in the general population. Given that there were two different datasets within the IMAGE dataset with different demographics, we corrected for this in two ways. First, in the adjusted model, we corrected for age and educational attainment in all four datasets. Additionally, to account for other factors that may differ between the case-control and the mathematical aptitude sub-datasets, we included a dummy variable called “study”, to account for the intrinsic differences in recruitment and other characteristics between the two sub-datasets. This is described under the “Statistical Analyses” section of the Methods. As can be seen from the ORs, and Figure 2B, the adjusted ORs are similar between the four cohorts. The CIs are wider in the IMAGE dataset because of the smaller sample size and the inclusion of an additional covariate for the sub-dataset. Further, the subsampling bootstrapping analyses does not identify any significant difference in ORs between IMAGE and both the subsampled C4 and MU datasets. Together, this suggests that the adjusted ORs estimated in the IMAGE datasets are robust.

Please note, we have now included a fifth cohort from LifeLines.

5. The heading 'rates of autism' seems misleading - my assumption is this should be 'rates of autism diagnoses' as described earlier. This term should be used consistently throughout as the media and non technical readers may pick this up incorrectly assuming it to mean autism rates (in adulthood autism population rates and rates of autism diagnosis are likely to be very different). Autism diagnoses were available from some of the studies only. Traits were obtained from the remainder (the subsequent subheading). The term autism diagnoses is later used in the same paragraph. This lack of precise terminology is unusual in this otherwise quite precisely drafted paper.

Thank you for pointing this out. We agree and have amended the manuscript accordingly. Where referring to our analyses and results, we have now used the term ‘rates of autism diagnosis’ throughout the manuscript. However, when introducing or referring to other work, we have used the term ‘rates of autism’ as autism was ascertained very differently in these studies, including, in some cases, through a research diagnosis.

6. Lack of comparability is mentioned further down and how it was handled using bootstrap sampling - given the heterogeneity of the data sources more (or clearer) information on how such differences were addressed would be helpful.

Thank you. We agree. In the methods, we have provided further details on how the datasets differ, and expand on the rationale behind the subsampling based bootstrap analyses.

7. Brain types is not explained in the methods (or indeed the introductions), which may be a rather specialist term of interest to the authors. And why is it relevant or necessary? Unless clearly justified it could be dropped?

We have now introduced the concept of Brain Types in the methods section. The addition of the Brain Types analyses is a natural extension of the empathizing-systemizing analyses we have conducted. It is needed to demonstrate that not only is there a shift in scores in the EQ-10 and the SQ-10, but this shift is also present when investigating the scores on the EQ-10 and SQ-10 relative to each other, i.e. their discrepancy.

Results

8. Findings in Figure 4a are not described in the text. There is not a description of associations of gender identity with autism (diagnoses/traits) compared with the other conditions studied. To the non-specialist eye, except for schizophrenia, there is no difference between disorder associations in figure 4. In figure 4a autism seems to stand out (we are not told is this traits or diagnoses).

We have now described this in the results, and clarified in the figure that these pertain to diagnosis rather than traits. In the C4 dataset, the ORs for autism, ADHD, bipolar disorder and depression were similar to each other. In comparison, the ORs for OCD and LD were about half that for autism. In contrast to the C4 dataset, in the MU dataset, the ORs for autism was the largest, followed by the two mood disorders (depression and bipolar disorder). Notably, the OR for depression was similar in both the C4 and the MU datasets.

9. Key concern.

If the association of gender is not specific to autism the article title should reflect this. And it would no longer be a surprising or original finding as transgender people are more likely to be distressed or mentally unwell (according to studies of people presenting for medical or surgical interventions). All of the neuropsychiatric conditions studied include mental distress (even intellectual disability). It is unsafe to presume that each is separately and accurately delineated using the methods described; a common distress dimension may possibly explain the underlying thrust of the study findings. Thus, a devil's advocate argument is that all this paper shows is that mental distress (of various kinds) is associated with declared transgender status. This would be old news, albeit on a more substantial scale (adding something to the literature but not a major finding). The paper does not convincingly show that the association with gender is specific to autism. It suggests therefore an already known non specific association of mental morbidity with transgender status (both self described). There may be opportunities here to clarify this issue or conduct further analyses addressing this issue. The study is unusual (and particularly good) in examining associations with a range of conditions in addition to autism (and not just autism alone).

The authors seem to touch on this issue but not tackle it head on in the discussion in the paragraph: "Our study also identified that transgender individuals have elevated rates of autism-related neurodevelopmental and psychiatric conditions in two datasets when compared to cisgender individuals". Interestingly they state that "In contrast to the ORs for autism, the ORs for these related conditions varied between the two datasets. While the OR for schizophrenia in transgender individuals was high in the C4 dataset, it was low and failed to reach statistical significance in the MU dataset." This argument touches on but does not tackle sufficiently my main concern, which could presumably be formally modelled. Their position on this question then seems to emerge in the final concluding statement and in a discussion comment on causality where they say that 'a few hypotheses may explain the over-representation of autism and related conditions in transgender individuals...' [note 'and']. If that is actually what this study shows that should be stated in the title (which as I said earlier seems to suggest that the over-representation is specific to autism). The discussion in this paragraph includes quite interesting causal explanations overlooking the more parsimonious explanation that underlying distress (however that may be described or characterised), explains the main finding, common to all these conditions.

Thank you for raising this important point of discussion and the “devil’s advocate argument” for us to carefully consider. We agree that this is an important area to further investigate. Within believe the best and most parsimonious way to investigate this is using a logistic regression, as follows:

Transgender vs cisgender $\sim \beta_1$ Autism + β_2 ADHD + β_3 Depression + β_4 OCD + β_5 Schizophrenia + β_6 Bipolar disorder + (β_7 LD) + β_8 Educational attainment + β_9 Age

We conducted this in C4 and in the MU datasets, but excluding LD in the MU dataset.

Our results are as follows:

	C4			MU		
	Beta	SE	p-value	Beta	SE	p-value
Autism	1.23	0.05	2.00E-16	1.37	0.18	2.77E-14
ADHD	1.13	0.06	2.00E-16	0.26	0.15	0.09
Bipolar	1.14	0.07	2.00E-16	0.35	0.19	0.07
Depression	1.27	0.04	2.00E-16	1.25	0.10	2.00E-16
LD	0.42	0.07	1.60E-08	NA	NA	NA
OCD	0.47	0.07	3.90E-10	0.19	0.21	0.37
Schizophre	0.67	0.12	3.20E-08	-0.11	0.52	0.83

As can be seen in the results, only autism and depression are significantly associated with gender identity in both datasets. Notably, in both datasets, autism and depression have the highest regression betas (and, as a consequence, highest ORs). Further, the estimates of the regression betas are similar across both datasets for depression and autism. This suggests that both general distress (likely represented here through depression) and early-onset

neurodevelopmental characteristics (i.e. autism diagnosis) are elevated in transgender and gender diverse individuals.

We expand on this in the results and discussion:

“To further clarify the role of autism compared to other neurodevelopmental and psychiatric conditions, we conducted multiple regression to investigate if the relative effects of autism on transgender and gender diverse identities compared to other neurodevelopmental and psychiatric conditions. In the C4 dataset, depression had the highest OR (OR = 3.55, 95%CI = 3.84 – 3.29, p-value < 2×10^{-16}) followed by autism (OR = 3.43, 95%CI = 3.79 – 3.11, p-value < 2×10^{-16}). In the MU dataset, we obtained very similar ORs. Autism had the highest OR (OR = 3.94, 95%CI = 5.61 – 2.77, p-value < 2×10^{-16}) followed by depression (OR = 3.50, 95%CI = 4.25 – 2.89, p-value < 2×10^{-16}). ORs for other conditions are provided in the **Supplementary Table 10.**”

“Studies have demonstrated elevated rates of several mental health conditions in transgender and gender diverse individuals. Multiple regression analyses demonstrate that whilst transgender and gender diverse individuals have elevated rates of multiple neurological and psychiatric conditions, the effect of autism higher than the effect of other neuropsychiatric conditions tested, with perhaps the exception of depression. These results suggest that transgender and gender diverse individuals have elevated rates of autism and a general distress component (largely, depression). Given that it is an adult sample, it is unclear how many individuals with other neurodevelopmental and psychiatric conditions have been misdiagnosed^{12,13} or have undiagnosed autism¹⁴. Epidemiologically relevant datasets with detailed clinical characterization are required to truly understand the unique association between autism diagnosis and gender identity above and beyond other psychiatric and neurodevelopmental conditions.”

10. An omission is any reference to the literature from transgender health clinics and intervention studies showing elevated levels of mental distress and disorder in such populations, i.e. arguably this is already known although this is a more thorough and substantial study of this question.

Thank you. We have now provided, in the introduction, details about studies investigating the association between gender identity and other mental health conditions and mental distress. Again, similar to autism, this is a heterogeneous field. There are a large number of studies finding an association between gender identity and depression, anxiety, distress, self-harm and suicide. There is also some literature on gender identity and substance abuse. However, our review of the literature identified that there were fewer studies investigating the association between gender identity and other neurodevelopmental and psychiatric conditions. Further, most of the studies have sample sizes in hundreds or early thousands. So, we believe that our study adds to the literature not just in terms of examining the odds of autism diagnosis in transgender and gender diverse individuals, but also in terms of examining the odds of selected neurodevelopmental and psychiatric conditions in transgender and gender diverse individuals.

Our amended introduction reads: “In parallel, studies have also investigated the rates of mental health conditions and mental distress in transgender and gender diverse individuals, including individuals with GD (e.g. refs²⁻⁹). The literature has been heterogeneous with varying research methodologies and sample sizes¹⁰. Two recent literature reviews have identified higher rates of mental health conditions and mental distress (notably depression, anxiety, and substance use disorders) in transgender and gender diverse individuals compared to cisgender individuals^{5,10}. Most of the research on mental health has focused on depression, substance misuse, and anxiety, with still limited research on other neurodevelopmental and psychiatric conditions. It is also unclear currently how the elevated rates of autism diagnosis in transgender and gender diverse individuals compare to other neurodevelopmental and psychiatric conditions. To our knowledge, barring one study¹¹, none of these studies have compared the rates of other related neurodevelopmental and psychiatric conditions in transgender and gender diverse individuals compared to cisgender individuals, making it difficult to estimate if the observed effects are specific to autism.”

Apart from this the paper describes possible biases and limitations in a balanced and transparent way. I would be very interested in seeing a revision addressing the concerns I have raised.

Thank you once again for reviewing this manuscript and the very helpful comments.

Reviewer:

Traolach Brugha, University of Leicester, 4 October, 2019.

Reviewer #4 (Remarks to the Author):

Thank you for the opportunity to review this paper. The topic of gender variation among individuals with autism is important to understand so that providers can better identify areas of benefit and need. Unfortunately, there are several limitations of the current study that prevent me from recommending publication at this time. I provided detailed feedback on the introduction and method section to help guide future revisions of the paper.

We thank you for reviewing the manuscript and providing us with very useful feedback. Please find our responses below.

Introduction

1. The introduction needs revision to provide focus and clear definition of terms. It would be helpful to start the paper with a brief discussion of gender identity as a continuous construct and the range of possible identities commonly identified in the literature. This overview would help the reader distinguish between the broader idea of gender variation (i.e., gender identity that does not always correspond to sex assigned at birth) from specific types of variations (e.g., transgender, non-binary, gender fluid, and gender queer). Along these same lines, it is not appropriate to include terms such as non-binary, gender fluid, and gender queer as examples of transgender identity since these terms represent individuals at different points on the sex/identity spectrum.

Thank you. We have now included lines in the text that clearly define gender identities. We have not used the term in a completely continuous manner (rather, it has multi-categorical components) as this may be challenged by some individuals' lived experiences (for example, it is unclear where agender individuals would fall on this continuity). In order to be inclusive and accurately reflect the samples ascertained in the different datasets, we have changed the term "transgender" to "transgender and gender diverse" as suggested by Reviewer #1. This terminology, we believe, is more representative of the diverse gender groups we have included and is pertinent to your final point about non-binary, gender-fluid and gender queer individuals as not being examples of a transgender identity.

The introduction would be strengthened if authors provided estimates of
(1) the number of gender variant and transgender people in the general population and
(2) the number of gender variant and transgender people among those with autism;
followed by

(1) The number of individuals with autism in the general population and
(2) the number of people with autism in the gender variant and transgender population.

We have now provided the number of transgender and gender diverse individuals in the general population and the number of autistic individuals in the general population. However, we have not provided numbers of individuals who are transgender and gender diverse and autistic as, as mentioned in the introduction, there are no large-scale studies that have provided un-biased and representative estimates.

Following, limitations of previous studies could be introduced, including small clinic-based samples, ascertainment of gender variation from one checklist item on the CBCL, and defining gender variation by endorsement of gender dysphoria. The details, strengths, and limitations of the datasets used in these analyses belong elsewhere in the paper (i.e., method and discussion).

Thank you. We agree and have moved this paragraph elsewhere in the manuscript (Methods, to provide an overview of the cohorts).

Method

Only one of the four datasets provided can answer the research questions with acceptable methods. The other three datasets have significant limitations that prevent interpretation of results. As mentioned previously, gender identity (e.g., cisgender, transgender, non-binary) is distinctly different from biological sex assigned at birth (i.e., male, female, intersex). In the Channel 4 and Music studies, participants were asked about sex but not about gender. Moreover, when asked about sex, participants were given two options that could be considered both sex and gender (i.e., male and female) and one option that could be considered only gender (transgender). In the physical health survey, participants were asked about sex and gender but the response options for gender were male, female, non-binary, and other. Therefore, the physical health survey data can only be used to estimate gender variance and/or non-binary status.

The genetics dataset is the only one included in the study that asked about sex and gender and had appropriate response options for both stem questions. I would therefore include only the genetics dataset in future analyses. It is my opinion that the cost of limiting the sample to one dataset is superseded by the benefits of sound methodology.

Thank you for the feedback. We acknowledge that there are limitations with the way gender was captured in two of the surveys. This limitation has been acknowledged throughout the manuscript, and we have addressed it using permutation analyses. Please allow us to expand on several points in detail.

First, based on the recommendation of the reviewers, we have changed the terminology to “transgender and gender diverse” throughout the manuscript, to more accurately describe and capture the heterogeneous gender identities included in our analyses.

Second, as a direct consequence of this, we believe that the APHS dataset (along with IMAGE) are highly appropriate for the analyses. Specifically, in the APHS, we ask participants for “Sex assigned at birth” and “Current gender identity”. Here, 33 individuals had discordant sex and gender information – indicating that they were transgender. We have provided this information in the manuscript. By asking about both gender and sex, and by additionally changing the terminology to “transgender and

gender diverse”, we believe that we are able to accurately describe and capture the diversity of gender in the APHS.

Third, to address this reviewer’s concern, we have now included an additional (fifth) dataset from the LifeLines cohort. This dataset consists of 37,975 individuals including 15,527 cisgender males, 22,375 cisgender females, and 53 transgender and gender diverse individuals. Gender identity was ascertained using one of five options:

- At birth I was registered as female and I am female
- At birth I was registered as male and I am male
- At birth I was registered as female, but I am male
- At birth I was registered as male, but I am female
- Different from the options above, namely... (followed by a free text box where participants explained).

This dataset characterizes current gender identity better than the C4 and MU dataset.

In addition, participants also provided information on their autism diagnosis: “Do you have an autism diagnosis?” followed by “In what year was this diagnosed”. A total of 439 individuals indicated that they had an autism diagnosis. All participants also completed the AQ-10, and the majority of the participants had information on education and age at the time of completion of the survey.

The relative advantage of additionally using this data are a better description of gender, and a different recruitment method which will have different biases from the previous datasets. In contrast, however, this dataset is limited in that it is the oldest dataset in terms of age of participants, and subsequently has the fewest number of transgender and gender diverse individuals of all the datasets as transgender and gender diversity is higher in younger participants¹⁵. Further, this dataset also has a healthy volunteer bias in that individuals with severe mental health conditions were not recruited into the LifeLines cohort.

Given the limited sample size of transgender and gender diverse individuals, we conducted post-hoc statistical power analysis to identify how much power we had for χ^2 tests, logistic regression, and linear regression. For χ^2 tests, logistic regression, the statistical power was greater than 0.50, but less than 0.8, with the highest statistical power obtained for transgender and gender diverse and female comparisons. However, for linear regression with autistic traits, we achieved greater than 0.8 power. We proceeded with the analysis despite the modest statistical power achieved as we will still be able to compare the effect direction observed in this dataset with those in other datasets.

As can be seen in the Results section, χ^2 test indicated that transgender and gender diverse individuals had elevated rates of autism compared to cisgender individuals: OR = 5.50, 95%CI = 1.60 – 16.60, p-value = 0.002. This was not statistically significant after accounting for age and educational attainment, primarily due to the low statistical power as provided in the Supplementary Note. However, we did observe concordant effect

direction: (LifeLines: OR = 3.03, 95% CI = 0.72 – 12.76, p-value = 0.13). Similarly, for the AQ-10, in the LifeLines dataset, transgender and gender diverse individuals scored higher than females (Beta = 1.23 ± 0.25 , p-value = 1.4×10^{-6}) and nominally higher than males (Beta = 0.51 ± 0.25 , p-value = 0.045).

Fourth, it is precisely to address this reviewer's concern about potential measurement error of gender, that we used a statistical bootstrapping approach to compare the Odds Ratios (ORs) between the two larger datasets (C4 and MU) and the two smaller datasets (and now a fifth, smaller dataset, LifeLines). Bootstrapping creates 10,000 datasets from the larger of the two cohorts that are matched in terms of the number of males, females and transgender and gender diverse individuals in the three smaller cohorts. If we identify similar ORs in these subsamples as identified in the smaller cohorts, then, this would suggest that the results are robust to the different characteristics of the cohorts including ascertainment of gender and autism diagnosis, age, recruitment methods. We do not find that the ORs are statistically any different across the five datasets using this approach. Thus, these results suggest that the results are statistically similar between the two larger cohorts and the three smaller cohorts. Specifically, this also suggests that the way gender was ascertained in the two larger cohorts does not statistically affect the results.

Fifth, we have directly tested the use of different terms to capture gender identity in the MU dataset ("Transgender" vs "Other"). Sensitivity analysis in the MU dataset did not identify differences in the rates of autism diagnosis between participants who indicated 'Other' vs 'Transgender'.

Sixth, we observe similar ORs for autism diagnosis in our study and in the largest study based on individuals recruited from GD clinics. For instance, the ORs identified in our datasets are: ORs: 3.03 - 6.36. In the largest study of children from a Gender Dysphoria clinic, the OR identified is 4.70 (95%CI: 2.89 – 7.65). Thus, the strength of the study is that regardless of how gender information was collected, ascertained or inferred, we identify similar ORs across the five datasets. That this OR is similar to that identified in a study investigating gender dysphoria only further strengthens our results. Therefore, the fact that we have observed similar quantitative results in different datasets suggests both replicability and generalizability of the findings. This has specifically been pointed out by Reviewer #3 who suggests that the magnitude of the samples suggests robustness of the results.

Tying all of these together, our assessment based on multiple sensitivity analyses, and robust, permutation analyses, is that the results in the larger datasets are not biased due to differences in how sex and gender was ascertained. In sum, despite the possibility of measurement error in the C4 and the MU datasets, which we have acknowledged throughout the manuscript, the quantitative findings are robust and similar across the five datasets. So, despite the differences in the datasets, the convergence of results across datasets of varying sample sizes suggests that these findings are valid, robust, and largely generalizable, as suggested by the other reviewers.

Reviewers' comments:

Reviewer #1 (Remarks to the Author):

The authors have been responsive to the first round reviewer comments. As mentioned in my previous review, this manuscript has several strengths, including, multiple survey types with converging evidence across sub-studies, a sophisticated approach to calculating ORs to account for differing (and small) sample sizes, and consideration of other co-occurring conditions and the effect size in relation to autism. There are a six remaining recommendations in the revision:

1. Abstract, final sentence: If there is room, consider broadening your statement. These finding may have clinical implications for tailoring supports and care to the subset of gender-diverse individuals who are autistic/neurodiverse. It is not just about access.

2. Introduction, paragraph 2: The Hilse-Gorman and Kaiser studies were not small sample sizes. Please adjust the first sentence of paragraph 2 to read, "many with small sample sizes" instead of, "with small sample sizes".

3. Introduction, paragraph 3 currently reads, "It is also likely that GD clinic-based samples may have ascertainment bias as they are likely to oversample individuals with mental health challenges." This sentence reveals limited awareness of how gender-diverse/transgender youth with gender dysphoria receive care. It incorrectly suggests that youth gender programs are focused on mental health and draw the least mentally-well youth. In fact, most youth gender care programs are focused primarily on supporting gender dysphoria, including with gender-affirming medical care. There is some evidence that it is the youth who are not able to access such care who present with the greatest mental health problems. Your sentence should be tempered. Consider this: In the Netherlands, the Center of Expertise on Gender Dysphoria was the only gender service for the entire country for many years, including during the time when the first paper on the co-occurrence was published. Yes, it is a clinical service, but it was the only clinical service for gender-diverse youth to receive medical gender care. The standard of care for transgender youth with gender dysphoria is evaluation (through a gender specialist program) and consideration for gender affirming supports. It is wrong to assume that transgender/gender-diverse youth with gender dysphoria who present to gender clinics are necessarily unrepresentative of that group of transgender youth, overall, who experience intense gender dysphoria. These are the youth with gender dysphoria who need care. At the very most you could say, "youth attending gender clinics likely represent the young people with 1) the most intense gender dysphoria, such that it warranted a referral, and 2) those youth who can access to this care (e.g., with more accepting parents, greater resources, etc.)" Please tone down the suggestion that they have more "mental health challenges" unless you link possible greater mental health challenges to these young people's more overt/strong gender dysphoria (which would have signaled the referral to the gender clinic.)

4. Introduction, page 5, last sentence currently reads, "Additionally, as an exploratory analysis, we investigated whether transgender and gender-diverse individuals are more likely to suspect that they have undiagnosed autism compared to cisgender individuals." This approach makes sense given the recent findings of under-diagnosis of autism in girls and women. Might you link this literature here?

5. A general comment about "Intersex": Intersex is used in some of the surveys you employed as an alternative to "male" and "female". Yet, the vast majority of people with differences of sex development (DSD)/intersex conditions are male or female sex and also intersex. Please consider adjusting your language in the opening of the introduction to reflect this, as the surveys you employ suggest that intersex people do not have a binary sex, yet most do.

6. The manuscript has a more human quality now with the addition of a paragraph about the lived experience of individuals with the co-occurrence and more sensitive language to gender-diversity throughout. It is still missing any statements about the rights of these individuals. Please note two important references that directly comment on the rights of autistic transgender individuals (listed below). In the first, which is a position statement of three major advocacy organizations, the rights of transgender gender-diverse people are emphasized, including the right to access gender-affirming care. The second reference is a Delphi study that presents the first consensus clinical guidelines for the care of autistic gender diverse adolescents. The international expert team agreed that autism should not preclude consideration for and access to gender affirming care for transgender autistic youth. These statements could be included in your Discussion.

Reference 1: the Joint Statement of the Autistic Self Advocacy Network, The National Center for Transgender Equality, and the National LGBTQ Taskforce on the rights of autistic gender-diverse people:

https://autisticadvocacy.org/wp-content/uploads/2016/06/joint_statement_trans_autistic_GNC_people.pdf

Reference 2: The Delphi consensus guidelines developed by clinical experts in the co-occurrence of autism and gender dysphoria/incongruence:

Strang, J.F., Meagher, H., Kenworthy, L., de Vries, A.L.C., Menvielle, E., Leibowitz, S., ... Anthony, L.G. (2016). Initial clinical guidelines for co-occurring autism spectrum disorder and gender dysphoria or incongruence in adolescents. *Journal of Clinical Child and Adolescent Psychology*, 47(1), 105-15. DOI: 10.1080/15374416.2016.1228462 PMID: 27775428

7. Finally, as a comment to the correspondence between the authors and Reviewer 4, the solution of expanding the language to "transgender and gender-diverse" does appear to cover the terminology issue with the surveys. In fact, the phrase "gender-diverse" could be used throughout as an umbrella that includes transgender. This could increase precision, but the current terminology used is perfectly in-line with standards for gender-diversity related reporting. Therefore, it may be best to leave it as is for clarity.

Reviewer #2 (Remarks to the Author):

I am largely satisfied with the authors responses to my concerns.

With respect of point

12. Line 466 to 468 Wouldn't a MANOVA be the more appropriate test of this hypothesis here, followed by the univariate analyses. As these are evaluations of gender differences, d-values need to be reported as estimates of effect size, not simply stating the extreme pvalue. We are unsure if MANOVA is the appropriate test here. An ANCOVA would be appropriate if we wanted to include autism diagnosis as a covariate. But we have done a regression (which is equivalent to an ANCOVA) just after the initial univariate analyses. We have also reported the Cohen's Ds in Supplementary Table 6 for all comparisons.

The point I was intending to get at is that AQ-10, SQ-10, EQ-10, and SPQ-10 are probably related, and a MANOVA or MANCOVA would assess these for the multivariate structure of the data, providing a better assessment of the differences between the two groups. However, this was not a major point, and I am happy to let this lie.

Mark A Stokes

Reviewer #3 (Remarks to the Author):

The paper as well as reporting associations should describe for the reader actual numbers of affected individuals in each group including numerators and denominators in respect of prevalence estimates. This point was raised previously and has not been addressed:

The introduction would be strengthened if authors provided estimates of

(1) the number of gender variant and transgender people in the general population and (2) the number of gender variant and transgender people among those with autism; followed by

(1) The number of individuals with autism in the general population and

(2) the number of people with autism in the gender variant and transgender population.

Now that my initial concerns have been (partly addressed) on re-reading this much improved revision of the original draft article I have a number of concerns about the study conclusions to raise that are now more apparent.

The discussion of explanatory hypotheses is narrow. A more parsimonious interpretation of the study results is that associations with many neuropsychiatric conditions have been found with relatively weak evidence that the findings tell the reader something that is specific to autism. Arguably the article title could be potentially misleading as it suggests much more specific findings for autism than the wider results show. The abstract goes some way to acknowledging this: transgender and gender-diverse individuals also had higher rates of six other neurodevelopmental and psychiatric diagnoses compared to cisgender individuals (ORs: 1.92 (learning disorders) to 6.39 (schizophrenia)). Nevertheless it reaches the narrower conclusion that across five independently recruited datasets, transgender and gender-diverse individuals have elevated rates of autism diagnosis and higher autistic traits, compared to cisgender individuals.

Similarly, the discussion of hypotheses considers autism issues and not explanations of a much broader more parsimonious and less specific kind in relation to neuropsychiatric disorders in general.

Lesser detailed points of advice:

The following claim is over stated (because validation studies based on case control designs (in contrast to the recommended cohort design) are known to over estimate sensitivity and specificity of validity estimates): 'All participants completed [four] well-validated, short self-report psychological measures: the Autism Spectrum Quotient-10 (AQ-10)⁴⁷, a measure of autistic traits;...'

Reference: Allison, C., Auyeung, 1076 B., Baron-Cohen, S., Bolton, P. F. & Brayne, C. Toward brief 'Red Flags' for autism screening: The Short Autism Spectrum Quotient and the Short Quantitative Checklist for Autism in toddlers in 1,000 cases and 3,000 controls [corrected]. *J. Am. Acad. Child Adolesc. Psychiatry* 51, 202– 1080 212.e7 (2012).

Articles cited for autism prevalence.

Two cited articles are surveillance (administrative) and not prevalence studies: Baio, J. et al. Prevalence of Autism Spectrum Disorder Among Children Aged 8 Years — Autism and Developmental Disabilities Monitoring Network, 11 Sites, United States, 2014. *MMWR. Surveill. Summ.* 67, 1–23 (2018). Xu, G., Strathearn, L., Liu, B. & Bao, W. Prevalence of Autism Spectrum Disorder Among US Children and Adolescents, 2014-2016. *JAMA* 319, 81–82 (2018).

There is an omission of a reference to the widely cited Baxter et al systematic review on the Global Burden of Diseases autism estimates (in Psychological Medicine) and of adult autism prevalence estimates based on general population probabilistic samples throughout England, using detailed autism examination procedures, not relying on clinical service or administrative data, by Brugha et al (Arch Gen Psychiat and Br J Psychiat).

T. Brugha, University of Leicester.

Reviewer #4 (Remarks to the Author):

I thank the authors for responding to my previous comments on this paper. The revised version of the manuscript is much improved and will make an important contribution to the literature. Additional revision would help focus the manuscript to highlight major points and important study results. Specific recommendations are outlined below.

Throughout the manuscript:

- It would be easier to read the paper if “transgender and gender diverse” were changed to “gender diverse” (which includes transgender).
- Please use person-first language throughout the manuscript (e.g., “have autism” rather than “be autistic” and “children and adolescents with autism” rather than “autistic children and adolescents”).
- Specify “cisgender males,” “cisgender females,” and “cisgender males and females” for clarity.

Title:

- The title would be easier to read if shortened; consider “Elevated rates of autism and other developmental and psychiatric diagnoses in gender-diverse individuals: A study of 5 datasets”

Introduction:

- Revise the definition of gender diverse to state “Individuals whose gender does not always correspond to the sex assigned at birth.”
- Delete the sentence on lines 86-88 that reads “Most of these were conducted in small samples, and have not investigated whether atypical sensory sensitivity (now a core feature of autism) is elevated in transgender and gender-diverse individuals.” The limitation of small sample sizes in previous studies has already been noted and there is no need to highlight a lack of information on sensory sensitivities over other autism symptoms as a limitation of previous research.
- Suggest revising the second aim of the study to read “elevated traits related to autism.”

Figure 1:

- Add Lifeline measures of autism traits.
- Replace description of figure, which can be described in text, with a list of abbreviations used in the figure.

Methods:

I believe the Methods section requires full revision for clarity and to help facilitate interpretation of study results. My specific suggestions are as follows:

- Consider the headings: Overview of datasets, Autism diagnoses, Gender identity, Autism traits; Other non-ASD diagnoses, and Ethics.

- The overview of datasets should be thorough yet succinct and include only relevant information (e.g., design, number of participants, age of participants, demographic characteristics of participants).
- The section on autism diagnoses should outline how autism diagnoses were defined in each dataset. Similarly, other sections should outline how gender identity, autism traits, and other non-ASD diagnoses were defined in each dataset. It would be helpful to include more information on the questionnaires used to measure autism traits.
- The first sentence of the methods section is better suited for the discussion section: "The five datasets were recruited differently, inevitably with different potential ascertainment biases; however, given this sampling heterogeneity, observing similar results across these datasets might suggest that the results are unlikely to be false positives."
- There is a typo in line 169.
- The analyses of brain types are better suited for another manuscript. The introduction does not address brain types and why these analyses would be important to understand autism traits among gender diverse individuals. The discussion also does not provide enough dialogue about the interpretation and importance of these results.

Results:

- The results of analyses pertaining to supplementary data could be significantly shortened. Otherwise these data detract from the main analyses. I suggest adding a few brief sentences to the methods section on how and why these analyses were conducted, and then reference the main findings and supplementary tables in the results section.

Discussion:

- Suggest revising the discussion to first address major findings related to primary aims outlined in the introduction and then major findings related to secondary aims.

Reviewers' comments:

Reviewer #1 (Remarks to the Author):

The authors have been responsive to the first round reviewer comments. As mentioned in my previous review, this manuscript has several strengths, including, multiple survey types with converging evidence across sub-studies, a sophisticated approach to calculating ORs to account for differing (and small) sample sizes, and consideration of other co-occurring conditions and the effect size in relation to autism. There are a six remaining recommendations in the revision:

Thank you very much for a detailed review of the manuscript. This has been a very helpful discussion, and we have attempted to address all your comments below.

1. Abstract, final sentence: If there is room, consider broadening your statement. These finding may have clinical implications for tailoring supports and care to the subset of gender-diverse individuals who are autistic/neurodiverse. It is not just about access.

Thank you, we have now modified our concluding line in the abstract to read as: "The results may have clinical implications for improving access to mental health diagnosis and tailoring subsequent support and care for transgender and gender-diverse individuals"

2. Introduction, paragraph 2: The Hilse-Gorman and Kaiser studies were not small sample sizes. Please adjust the first sentence of paragraph 2 to read, "many with small sample sizes" instead of, "with small sample sizes".

We agree. We have now amended this to read as follows: "A few studies, mostly clinic-based, typically with small sample sizes, and in individuals with gender dysphoria (GD, defined as persistent distress arising from a mismatch between sex assigned at birth and gender identity), have investigated the link between autism/autistic traits and gender diversity." Please note the inclusion of the word 'typically' to convey that these studies are largely but not always conducted using small sample sizes.

3. Introduction, paragraph 3 currently reads, "It is also likely that GD clinic-based samples may have ascertainment bias as they are likely to oversample individuals with mental health challenges." This sentence reveals limited awareness of how gender-diverse/transgender youth with gender dysphoria receive care. It incorrectly suggests that youth gender programs are focused on mental health and draw the least mentally-well youth. In fact, most youth gender care programs are focused primarily on supporting gender dysphoria, including with gender-affirming medical care. There is some evidence that it is the youth who are not able to access such care who present with the greatest mental health problems. Your sentence should be tempered. Consider this: In the Netherlands, the Center of Expertise on Gender Dysphoria was the only gender service for the entire country for many years, including during the time when the first paper on the co-occurrence was published. Yes, it is a clinical service, but it was the only clinical service for gender-diverse youth to receive medical gender care. The standard of care for transgender youth with gender dysphoria is evaluation (through a gender specialist program) and consideration for gender affirming supports. It is wrong to assume that transgender/gender-diverse youth with gender dysphoria who present to gender clinics

are necessarily unrepresentative of that group of transgender youth, overall, who experience intense gender dysphoria. These are the youth with gender dysphoria who need care. At the very most you could say, “youth attending gender clinics likely represent the young people with 1) the most intense gender dysphoria, such that it warranted a referral, and 2) those youth who can access to this care (e.g., with more accepting parents, greater resources, etc.)” Please tone down the suggestion that they have more “mental health challenges” unless you link possible greater mental health challenges to these young people’s more overt/strong gender dysphoria (which would have signaled the referral to the gender clinic.)

Thank you for raising this important point. We completely agree with this assessment. We have now amended the statement to read as follows: “It is also likely that young people attending GD clinics represent young people with the most intense gender dysphoria, such that it warrants a referral for clinical care, and/or those young people who can access this care (e.g., with parents who are more tolerant of difference, or who have greater resources, etc.)”

4. Introduction, page 5, last sentence currently reads, “Additionally, as an exploratory analysis, we investigated whether transgender and gender-diverse individuals are more likely to suspect that they have undiagnosed autism compared to cisgender individuals.” This approach makes sense given the recent findings of under-diagnosis of autism in girls and women. Might you link this literature here?

We have now included this and provided references. The modified lines now read: “Finally, whilst the previous literature has provided compelling evidence that autism is under-diagnosed (or mis-diagnosed as other conditions) in cisgender females, it is unclear if this is true of transgender and gender-diverse individuals^{48–50}.”

We have provided the following references:

1. Lai, M.-C., Lombardo, M. V., Auyeung, B., Chakrabarti, B. & Baron-Cohen, S. Sex/gender differences and autism: setting the scene for future research. *J. Am. Acad. Child Adolesc. Psychiatry* 54, 11–24 (2015).

2. Loomes, R., Hull, L. & Mandy, W. P. L. What Is the Male-to-Female Ratio in Autism Spectrum Disorder? A Systematic Review and Meta-Analysis. *J. Am. Acad. Child Adolesc. Psychiatry* 56, 466–474 (2017).

3. Lai, M.-C. & Szatmari, P. Sex and gender impacts on the behavioural presentation and recognition of autism. *Curr. Opin. Psychol.* doi:10.1097/YCO.0000000000000575

5. A general comment about “Intersex”: Intersex is used in some of the surveys you employed as an alternative to “male” and “female”. Yet, the vast majority of people with differences of sex development (DSD)/intersex conditions are male or female sex and also intersex. Please consider adjusting your language in the opening of the introduction to reflect this, as the surveys you employ suggest that intersex people do not have a binary sex, yet most do.

We agree. We have attempted to provide more nuance to this by qualifying our statements. The amended sentences now read: “Gender identity is different from sex assigned at birth, which is typically classified as male or female primarily based on genitalia. Some individuals are born with chromosomal, genital, or hormonal sex-characteristics which vary from the male-female binary (intersex individuals) and who may be assigned as or raised as males or females.”

6. The manuscript has a more human quality now with the addition of a paragraph about the lived experience of individuals with the co-occurrence and more sensitive language to gender-diversity throughout. It is still missing any statements about the rights of these individuals. Please note two important references that directly comment on the rights of autistic transgender individuals (listed below). In the first, which is a position statement of three major advocacy organizations, the rights of transgender gender-diverse people are emphasized, including the right to access gender-affirming care. The second reference is a Delphi study that presents the first consensus clinical guidelines for the care of autistic gender diverse adolescents. The international expert team agreed that autism should not preclude consideration for and access to gender affirming care for transgender autistic youth. These statements could be included in your Discussion.

Reference 1: the Joint Statement of the Autistic Self Advocacy Network, The National Center for Transgender Equality, and the National LGBTQ Taskforce on the rights of autistic gender-diverse people:

https://autisticadvocacy.org/wp-content/uploads/2016/06/joint_statement_trans_autistic_GNC_people.pdf

Reference 2: The Delphi consensus guidelines developed by clinical experts in the co-occurrence of autism and gender dysphoria/incongruence:

Strang, J.F., Meagher, H., Kenworthy, L., de Vries, A.L.C., Menvielle, E., Leibowitz, S., ... Anthony, L.G. (2016). Initial clinical guidelines for co-occurring autism spectrum disorder and gender dysphoria or incongruence in adolescents. *Journal of Clinical Child and Adolescent Psychology*, 47(1), 105-15. DOI: 10.1080/15374416.2016.1228462 PMID: 27775428

Thank you. We agree. We have now modified the paragraph and included the two references. The new paragraph is as follows: “These findings must be interpreted in light of the lived experiences, rights, and clinical and daily life needs of transgender and gender-diverse individuals. Both autistic individuals and transgender and gender-diverse individuals are marginalized groups where the currently available support and understanding is inadequate⁷⁶. Both groups are also more likely than others to engage in self-harm, suicidal ideation and suicidal behaviors, and to have other vulnerabilities^{74,77-79}. This intersection of autism and gender diversity can be doubly distressing if adequate safe-guarding and support are not provided. A recent study demonstrated that a third of autistic individuals had their gender identity questioned because they were autistic⁷⁶. There is a need to ensure that autistic transgender and gender-diverse individuals have the right to express their gender, live with dignity, and receive social and legal recognition of their gender⁸⁰ (also see: https://autisticadvocacy.org/wp-content/uploads/2016/06/joint_statement_trans_autistic_GNC_people.pdf). Additionally, recent studies demonstrate that autistic characteristics partly differ between cisgender males and cisgender females^{50,81,82}. However, it is still unclear if autistic characteristics differ in transgender and gender-diverse individuals compared to cisgender individuals.

This co-occurrence requires gender-informed and neurodiversity-informed clinical care for autistic transgender and gender-diverse individuals.”

7. Finally, as a comment to the correspondence between the authors and Reviewer 4, the solution of expanding the language to "transgender and gender-diverse" does appear to cover the terminology issue with the surveys. In fact, the phrase "gender-diverse" could be used throughout as an umbrella that includes transgender. This could increase precision, but the current terminology used is perfectly in-line with standards for gender-diversity related reporting. Therefore, it may be best to leave it as is for clarity.

Thank you. We agree.

Reviewer #2 (Remarks to the Author):

I am largely satisfied with the authors' responses to my concerns.

We thank you for the very thorough review of the manuscript, and for the positive appraisal of the revised manuscript.

With respect of point

12. Line 466 to 468 Wouldn't a MANOVA be the more appropriate test of this hypothesis here, followed by the univariate analyses. As these are evaluations of gender differences, d-values need to be reported as estimates of effect size, not simply stating the extreme pvalue. We are unsure if MANOVA is the appropriate test here. An ANCOVA would be appropriate if we wanted to include autism diagnosis as a covariate. But we have done a regression (which is equivalent to an ANCOVA) just after the initial univariate analyses. We have also reported the Cohen's Ds in Supplementary Table 6 for all comparisons.

The point I was intending to get at is that AQ-10, SQ-10, EQ-10, and SPQ-10 are probably related, and a MANOVA or MANCOVA would assess these for the multivariate structure of the data, providing a better assessment of the differences between the two groups. However, this was not a major point, and I am happy to let this lie.

Thank you. Apologies that we had misinterpreted the comment. We agree that this would be a useful analysis to conduct, and we need to investigate these traits related to autism in line with their multivariate structure. It is to (partly) address this that we conducted the D-score and 'Brain Types' analyses as this investigates the discrepancy between the SQ and the EQ which are only weakly correlated with each other. Whilst we appreciate the suggestion, after some discussion, we have decided not to conduct further analysis primarily because of the complexity of the current manuscript. This will be a useful follow up analysis for future studies.

Reviewer #3 (Remarks to the Author):

The paper as well as reporting associations should describe for the reader actual numbers of affected individuals in each group including numerators and denominators in respect of prevalence estimates. This point was raised previously and has not been addressed:

The introduction would be strengthened if authors provided estimates of

- (1) the number of gender variant and transgender people in the general population and
- (2) the number of gender variant and transgender people among those with autism; followed by
- (3) The number of individuals with autism in the general population and
- (4) the number of people with autism in the gender variant and transgender population.

Apologies for missing out on providing all the details in the previous revision. Given that there are numerous studies that have used very different methods to provide estimates, we have provided approximate percentages based on these studies. These numbers vary widely, and it's difficult to provide numerator and denominator for each of these studies individually. Specifically, this is difficult for #3 and #4, as the numbers included in the study are small (quite often less than 100). Further, it is easier to estimate the number of transgender and gender diverse individuals and the number of autistic individuals in the general population than it is to estimate the number of autistic transgender and gender diverse individuals due to the paucity of large studies in adults where transgender and gender diversity is well defined (as opposed to individuals from gender dysphoria clinics).

Please see our responses to each of the points below:

(1) the number of gender variant and transgender people in the general population:

This is provided in the final line of the first paragraph: "Currently, 0.4 - 1.3% of the general population is estimated to be transgender and gender-diverse, although the numbers vary considerably based on how the terms are defined."

(2) The number of individuals with autism in the general population:

This is provided in the first paragraph: "Approximately 1 - 2% of the general population is estimated to be autistic based on a number of different methods, although these numbers vary between countries, age at the time of assessment and other criteria²⁻⁸."

3) the number of gender variant and transgender people among those with autism:

We have now endeavoured to provide this in the introduction. We have limited ourselves to studies that have investigated using a sample size of at least 1,000 individuals as these estimates are most reliable. This leaves us with only 4 studies, all of them conducted in children and adolescents (Janssen et al., 2016; May et al., 2017; Strang et al., 2014; and Hilse-Gorman et al., 2019). The first three studies used CBCL, and the final study used medical records and outpatient information about GD clinics. The % of transgender and gender-diverse identity in autistic and non-autistic individuals are similar in the first three, but different in the final study possibly due to the different

methods used. However, the relative rates are similar in all 4 (~ 7%). This is provided in the table below:

Relative risk	Percentages (autistic vs non-autistic)	PMID	Study name	Age	Sample-size	Measure
7.28	5.1% vs 0.7%	2886 1527	Janssen et al., 2016	6 yrs - 18 yrs	492 autistic; 1605 non-autistic	CBCL sex item 110
5.71	4% vs 0.7%	3056 1911	May et al., 2017	6 yrs - 18 yrs	176 autistic; 1605 non-autistic	CBCL sex item 110
7.71	5.4% vs 0.7%	2461 9651	Strang et al., 2014	6 yrs - 18 yrs	147 autistic; 1770 non-autistic	CBCL sex item 110
7	0.07% vs 0.01%	3092 0347	Hilse-Gorman et al., 2019	6 yrs - 18 yrs	48,762 autistic; 243810 non-autistic	Outpatient visit for GD

Please note, no adult study met our criteria for sample size. These, whilst cited in the introduction, have not been discussed in the manuscript due to this reason.

We have subsequently modified our introduction as follows:

“These studies have identified increased rates of gender diversity in autistic children and adolescents¹⁴⁻¹⁸, and adults^{19,20}, compared to the general population. Most of these studies in children and adolescents have used a single item on the Child Behavior Checklist (CBCL), a caregiver-report measure for behavioural problems, to quantify gender variance, and these have identified between 4% to 5.4% of autistic children may be transgender or gender-diverse compared to 0.7% of non-autistic children¹⁴⁻¹⁶. The largest of these, conducted in nearly 300,000 children, identified a four-fold likelihood of GD in autistic compared to non-autistic children (i.e., 0.07% of autistic children and 0.01% of non-autistic children)¹⁷. Despite the differences in percentages of transgender and gender-diverse identity identities in the studies using CBCL and GD information, the relative rates are largely similar (between 5.7 to 7.7).”

4)The number of people with autism in the gender variant and transgender population

We have now endeavoured to provide this information. There are five studies that we identified that have provided this information, and all of them have studied individuals presenting to GD clinics. The sample sizes are small (N = 39, 47, 204, 532, and 540). None of these studies have used the same methods to investigate rates of autism diagnoses in individuals not presenting to GD clinics, thus they have not provided the relative rates (and, consequently, aren't able to investigate if autism is statistically overrepresented in individuals presenting to GD clinics). The details of the study are:

Relative risk	Percentages (autistic vs non-autistic)	PMID	Study name	Age	Sample-size	Measure
NA	6.40%	2009 4764	de Vries et al.,	under 18	204	DISCO;

			2010			
NA	26%	2587 3995	Kaltiala-Heino et al., 2015	Under 18	47	co-occurring diagnoses
NA	23.10%	2665 1183	Shumer et al., 2016	Under 20	39	Asperger Syndrome Diagnostic Scale
NA	6.02% (certain), 6.39% (probable)	2942 7119	Heylens et al., 2018	Adults	532	Clinical chart data review
NA	4.80%	3059 6151	Cheung et al., 2018	Adults	540	Medical records

Based on this, we have updated our introduction: “A second set of studies has investigated rates of autism in both children and adolescents^{21–23} and adults^{24,25} with GD. These studies have identified that between 4.8% to 26% of individuals who present at GD clinics have an autism diagnosis based on several different criteria. The largest of these studies (N = 532²⁴, and N = 540²⁵) identified that 6.0% and 4.8% respectively of these individuals are autistic based on review of clinical and medical records. Though none of these studies have used a match control sample to investigate the relative rates of autism diagnoses, using a population estimate of 1 – 2%^{2–8} suggests that autism diagnoses are elevated in individuals presenting at GD clinics.”

Now that my initial concerns have been (partly addressed) on re-reading this much improved revision of the original draft article I have a number of concerns about the study conclusions to raise that are now more apparent.

The discussion of explanatory hypotheses is narrow. A more parsimonious interpretation of the study results is that associations with many neuropsychiatric conditions have been found with relatively weak evidence that the findings tell the reader something that is specific to autism. Arguably the article title could be potentially misleading as it suggests much more specific findings for autism than the wider results show. The abstract goes some way to acknowledging this: transgender and gender-diverse individuals also had higher rates of six other neurodevelopmental and psychiatric diagnoses compared to cisgender individuals (ORs: 1.92 (learning disorders) to 6.39 (schizophrenia)). Nevertheless it reaches the narrower conclusion that across five independently recruited datasets, transgender and gender-diverse individuals have elevated rates of autism diagnosis and higher autistic traits, compared to cisgender individuals.

Similarly, the discussion of hypotheses considers autism issues and not explanations of a much broader more parsimonious and less specific kind in relation to neuropsychiatric disorders in general.

Thank you for the comments. This is an important point that we’d like to address in terms of the motivation for pursuing the study, the results identified, the strengths of these results, and our subsequent interpretation.

First, the entire study was conducted within the framework of autism right from the beginning. Of the three primary aims, two pertain to autism (diagnosed autism (Aim 1), and traits related to autism (Aim 3)), and our secondary, exploratory analyses also pertain to autism (suspected autism). The other aim (Aim 2) is largely to provide further context to the presented results - i.e. is this specific to autism or is this more widely observed across other, related conditions? We agree that there is some non-specificity, but we'd like to think of it in terms of degree rather than as a binary. In other words, *what proportion* of the elevated rates of autism diagnoses is due to an increase in a liability for psychiatric and neurodevelopmental conditions more generally, and *what proportion* is specific to autism? This is a complex question, which cannot be addressed in considerable detail in this study.

Tied to this, five different cohorts have been used to test Aim 1, three for Aim 3 and *only* 2 for Aim 2. Hence, there is a discrepancy in the number of datasets used to test the association with autism, and the number used to test the association with other psychiatric and neurodevelopmental conditions. This is particularly important given that sex-assigned-at birth and gender were not separately collected in the MU and the C4 datasets. Thus, whilst these five different datasets provide confidence in our results about autism, we are relatively less confident about the results regarding other conditions as they have been validated only within two datasets. Our permutation analyses were conducted only for autism, and there are discrepant results between the MU and C4 datasets for the other conditions tested, which we address in our discussion.

Additionally, our analyses of other conditions are not exhaustive. We have tested only six other psychiatric and neurodevelopmental conditions, and this is, by no means, comprehensive. This is in contrast to our main analyses regarding autism, which we believe is substantially more comprehensive and robust as we have: 1. Tested this in five datasets; 2. Demonstrated using permutation analyses the convergence in effect size across the five datasets; 3. Supported the results using autistic traits and suspected undiagnosed autism analyses.

In sum, whilst we agree that there is a degree of non-specificity between gender identity and autism diagnosis, we do not think it is all non-specific. We believe that the degree of non-specificity is important, and the next step is to partition the elevated rates of autism observed in transgender and gender-diverse individuals into 'autism specific' and 'pan-psychiatry'. We have attempted to investigate this to an extent using multiple regression, but our study is not designed to investigate this comprehensively due to the limitations of the data available.

Keeping these considerations in mind, we have revised our manuscript based on your thoughtful suggestions here in the following manner.

Title:

Our current title reads as follows: "Elevated rates of autism and other neurodevelopmental and psychiatric diagnoses, and traits related to autism in transgender and gender-diverse individuals: a study of five datasets".

We think that this title reflects the balance of analyses conducted, keeping in mind the robustness of each separate analysis. We would like to highlight autism

diagnoses and autistic traits in line with the reasoning above, but don't want to omit that one of our aims was also to investigate other related conditions. Spelling out the other conditions investigated (or combining autism and the other conditions tested into 'neurodevelopmental and psychiatric conditions') would be disingenuous and misleading given the points raised above.

Abstract:

We agree that the conclusions in the abstracts are narrower than they should be. We have amended this to read as follows: "Rates of other neurodevelopmental and psychiatric conditions may also be elevated. The results may have clinical implications for improving access to mental health diagnosis and tailoring subsequent support and care for transgender and gender-diverse individuals."

Discussion:

We have now re-written paragraphs to discuss the findings in other conditions to ensure that the results aren't misinterpreted to imply specificity.

"However, this association with gender identity is not specific to autism. In two datasets, transgender and gender-diverse individuals had elevated rates of ADHD, bipolar disorder, depression, and OCD, learning disorders, and schizophrenia. compared to cisgender individuals. In one of the two datasets, we tested and confirmed that transgender and gender-diverse individuals had higher rates of LD compared to cisgender individuals. In the C4 dataset, we identified elevated rates for schizophrenia in transgender and gender-diverse individuals compared to cisgender individuals but were unable to replicate this in the MU dataset.

Our multiple regression analyses helped clarify the relative association strengths of these conditions with transgender and gender-diverse individuals. In both the MU and the C4 datasets, autism and depression had the highest effect sizes. Notably, in the MU dataset, none of the other conditions were significantly elevated in transgender and gender-diverse individuals after controlling for autism and depression, which is discordant with the results identified in the C4 datasets. This discrepancy in the results may be due to differences in sample sizes, ascertainment, or other cohort characteristics. For instance, the C4 study directly recruited participants to an autism study. This may oversample individuals with other co-occurring mental health conditions. In contrast, the MU dataset is a convenience sample collected over many months. There is some evidence to suggest that individuals with elevated genetic liability for schizophrenia, ADHD, and depression may be less likely to participate in studies^{62,63}, and, as a result, they may be underrepresented in the MU dataset. In addition, most of the participants in the C4 are from the UK, whilst most of the MU participants are from the US. Differences in diagnostic practices may also contribute to sampling differences. A more comprehensive investigation of the relative rates of neurodevelopmental and psychiatric conditions in transgender and gender-diverse individuals compared to cisgender individuals is warranted."

Additionally, we also amended our discussion to update new hypotheses: "Whilst our study does not test causality, a few hypotheses may explain the over-representation of autism and other neurodevelopmental and psychiatric conditions in transgender and gender-diverse individuals. First, autistic individuals may conform less to societal norms compared to non-autistic individuals, which may partly explain why a greater number of

autistic individuals identify outside the stereotypical gender binary. Second, prenatal mechanisms (e.g., sex steroid hormones) shaping brain development have been shown to contribute to both autism (and associated neurodevelopmental conditions) and gender role behaviour⁶⁷⁻⁷¹. It is unclear if prenatal sex steroid hormones also contribute to gender identity and this should be investigated in future studies. Neurodevelopmental conditions such as ADHD and learning disorders frequently co-occur with autism⁴⁷, and genetic evidence suggests a shared underlying liability for many of the co-occurring neurodevelopmental and psychiatric conditions^{72,73}. Finally, an alternative but not mutually exclusive explanation is that transgender and gender-diverse individuals have elevated vulnerabilities for multiple psychiatric challenges related to stressful life experiences in the contexts of unfriendly environments, discrimination, abuse and victimisation, explaining the elevated rates of mental health diagnoses^{74,75} .”

Lesser detailed points of advice:

The following claim is over-stated (because validation studies based on case control designs (in contrast to the recommended cohort design) are known to over estimate sensitivity and specificity of validity estimates): 'All participants completed [four] well-validated, short self-report psychological measures: the Autism Spectrum Quotient-10 (AQ-10)⁴⁷, a measure of autistic traits;...'

Reference: Allison, C., Auyeung, 1076 B., Baron-Cohen, S., Bolton, P. F. & Brayne, C. Toward brief 'Red Flags' for autism screening: The Short Autism Spectrum Quotient and the Short Quantitative Checklist for Autism in toddlers in 1,000 cases and 3,000 controls [corrected]. *J. Am. Acad. Child Adolesc. Psychiatry* 51, 202– 1080 212.e7 (2012).

We have now removed the term “well-validated”.

Articles cited for autism prevalence.

Two cited articles are surveillance (administrative) and not prevalence studies: Baio, J. et al. Prevalence of Autism Spectrum Disorder Among Children Aged 8 Years — Autism and Developmental Disabilities Monitoring Network, 11 Sites, United States, 2014. *MMWR. Surveill. Summ.* 67, 1–23 (2018). Xu, G., Strathearn, L., Liu, B. & Bao, W. Prevalence of Autism Spectrum Disorder Among US Children and Adolescents, 2014-2016. *JAMA* 319, 81–82 (2018).

There is an omission of a reference to the widely cited Baxter et al systematic review on the Global Burden of Diseases autism estimates (in *Psychological Medicine*) and of adult autism prevalence estimates based on general population probabilistic samples throughout England, using detailed autism examination procedures, not relying on clinical service or administrative data, by Brugha et al (*Arch Gen Psychiat* and *Br J Psychiat*).

Thank you. We have now cited the previously omitted articles. Additionally, we have amended the statement to reflect that not all studies were prevalence studies (using the term 'large-scale prevalence and surveillance studies'). The sentence now reads: “Approximately 1 - 2% of the population is estimated to be autistic based on large-scale prevalence and surveillance studies, although these numbers vary between countries, age at the time of assessment and other criteria²⁻⁸.”

T. Brugha, University of Leicester.

Reviewer #4 (Remarks to the Author):

I thank the authors for responding to my previous comments on this paper. The revised version of the manuscript is much improved and will make an important contribution to the literature. Additional revision would help focus the manuscript to highlight major points and important study results. Specific recommendations are outlined below.

Thank you for the positive appraisal and detailed feedback of the revised manuscript.

Throughout the manuscript:

- It would be easier to read the paper if “transgender and gender diverse” were changed to “gender diverse” (which includes transgender).

Thank you for the comment. We appreciate the reviewer’s suggestion. After careful consideration, we are inclined to not alter the terminology for several reasons (also considering Reviewer 1’s suggestion [final point; please see above]).

First, we are aware that there are differences in how the terminology is used. For instance, both the GLAAD “Glossary of Terms” (<https://www.glaad.org/reference/transgender>), and the APA (<https://www.apa.org/monitor/2018/09/ce-corner-glossary>) use the term ‘transgender’ to refer to anyone whose gender is different from their sex-assigned at birth. So, in some instances, the term transgender may suffice to describe the group of individuals included in this study. However, we note that the term ‘gender-diverse’ adds further clarity as some individuals may identify as ‘gender-diverse’ but ‘not transgender’. Second, in many of our surveys, we included a ‘transgender’ and an ‘Other’ option, so we believe that omitting the term transgender will not reflect the questions asked in the surveys. Third, the Royal College of Psychiatrists in the UK use the term ‘transgender and gender diverse’, (https://www.rcpsych.ac.uk/pdf/PS02_18.pdf), suggesting that this is very much in line with the current thinking as pointed out by Reviewer #1 (point 7). Finally, we would rather be more inclusive with our terms than omit specific terms, so we are keen to retain ‘transgender and gender-diverse’ throughout the manuscript.

- Please use person-first language throughout the manuscript (e.g., “have autism” rather than “be autistic” and “children and adolescents with autism” rather than “autistic children and adolescents”).

Whilst some autistic individuals may prefer a person-first language, an overwhelming number of autistic individuals prefer identity-first language. For instance, please see the work done by Kenny and colleagues in 2016 (<https://journals.sagepub.com/doi/pdf/10.1177/1362361315588200>). In light of this and our own experience of working with autistic individuals, and to be respectful, we would like to use identity-first language in this manuscript.

- Specify “cisgender males,” “cisgender females,” and “cisgender males and females” for clarity.

Thank you, we have updated this throughout the manuscript.

Title:

- The title would be easier to read if shortened; consider “Elevated rates of autism and other developmental and psychiatric diagnoses in gender-diverse individuals: A study of 5 datasets”

Please see our response to the point earlier about using the term ‘gender-diverse’ in lieu of ‘transgender and gender-diverse’. We have further amended the title to reflect the points raised by Reviewer #3. Our revised title now reads: “Elevated rates of autism and other neurodevelopmental and psychiatric diagnoses, and traits related to autism in transgender and gender-diverse individuals: a study of five datasets.”

Introduction:

- Revise the definition of gender diverse to state “Individuals whose gender does not always correspond to the sex assigned at birth.”

Thank you. We have retained the term ‘transgender and gender diverse’, but have now included the word ‘does not *always*’ in the definition as the term also includes genderfluid individuals whose gender may, at times, correspond to their sex assigned at birth.

- Delete the sentence on lines 86-88 that reads “Most of these were conducted in small samples, and have not investigated whether atypical sensory sensitivity (now a core feature of autism) is elevated in transgender and gender-diverse individuals.” The limitation of small sample sizes in previous studies has already been noted and there is no need to highlight a lack of information on sensory sensitivities over other autism symptoms as a limitation of previous research.

Thank you. This has been corrected.

- Suggest revising the second aim of the study to read “elevated traits related to autism.”

We have now made a distinction between autistic traits and traits related to autism. Autistic traits measure the variation in core autism features in the general population (e.g. AQ-10), and traits related to autism refer to a number of specific features in which autistic individuals differ, on average, from the general population (e.g. self-reported empathy, systemizing, sensory difficulties), which represent aspects of autistic traits. We have accordingly made changes in the text.

Figure 1:

- Add Lifeline measures of autism traits.

Thank you, we have now updated this to include the measure of autistic traits in LifeLines.

- Replace description of figure, which can be described in text, with a list of abbreviations used in the figure.

We believe the description will only aid the readers to understand the figures. So, we would like to retain the description and the list of abbreviations. We have now provided the list of abbreviations.

Methods:

I believe the Methods section requires full revision for clarity and to help facilitate interpretation of study results. My specific suggestions are as follows:

- Consider the headings: Overview of datasets, Autism diagnoses, Gender identity, Autism traits; Other non-ASD diagnoses, and Ethics.
- The overview of datasets should be thorough yet succinct and include only relevant information (e.g., design, number of participants, age of participants, demographic characteristics of participants).
- The section on autism diagnoses should outline how autism diagnoses were defined in each dataset. Similarly, other sections should outline how gender identity, autism traits, and other non-ASD diagnoses were defined in each dataset. It would be helpful to include more information on the questionnaires used to measure autism traits.

Thank you for this feedback. We agree that there is quite a lot of information and it can be better presented. To facilitate this, we maintain the overall structure by separating information based on dataset first. This is because each dataset is different and has several idiosyncratic features and separating the information based on the format the reviewer has suggested may make it even more challenging for the readers to understand. However, within description of each dataset, we have separated the information based on the following subheadings, based on the reviewer's suggestion:

1. Overview of the dataset
2. Ascertainment of gender identity
3. Ascertainment of autism diagnosis (and other diagnosis in the case of C4 and MU datasets)
4. Measuring traits related to autism

We have provided extra information where needed.

- The first sentence of the methods section is better suited for the discussion section: "The five datasets were recruited differently, inevitably with different potential ascertainment biases; however, given this sampling heterogeneity, observing similar results across these datasets might suggest that the results are unlikely to be false positives."

We agree. This is also repeated in the discussion, so we've removed this sentence.

- There is a typo in line 169.

Thank you, this has now been corrected.

- The analyses of brain types are better suited for another manuscript. The introduction does not address brain types and why these analyses would be important to understand autism traits among gender diverse individuals. The discussion also does not provide enough dialogue about the interpretation and importance of these results.

The addition of the Brain Types analyses is a natural extension of the empathizing-systemizing analyses we have conducted (e.g, please see: <https://www.pnas.org/content/115/48/12152>). It is needed to demonstrate that not only is there a shift in scores in the EQ-10 and the SQ-10, but this shift is also present when investigating the scores on the EQ-10 and SQ-10 relative to each other, i.e. their discrepancy, which is critical information that cannot be derived from the EQ-10 and SQ-10 analyses alone, which requires the Brain Types analyses to investigate.

We have now provided further information about this in the results section: "The previous analyses investigated the association between gender identity and traits related to autism individually. We next investigated if there are differences in relative scores on the EQ-10 compared to the SQ-10 in the three gender categories using 'Brain Types'."

And in the discussion section: "Importantly, these effects were also observed when investigating the discrepancy of scores on the EQ-10 and SQ-10 using the 'Brain Types' analyses."

Results:

- The results of analyses pertaining to supplementary data could be significantly shortened. Otherwise these data detract from the main analyses. I suggest adding a few brief sentences to the methods section on how and why these analyses were conducted, and then reference the main findings and supplementary tables in the results section.

Thank you for the suggestion. We agree that this is a long paper, and the results section has a lot of details. However, after re-reading the results section a few times, we realized that further reduction would only confuse the readers by providing them with incomplete information. Further, as these results are discussed in detail in the discussion section, we want to ensure that the readers can follow even if they do not read the supplementary information.

Discussion:

- Suggest revising the discussion to first address major findings related to primary aims outlined in the introduction and then major findings related to secondary aims.

Thank you. We have now restructured our discussion to first provide an overview, followed by discussing each one of the primary aims and then the secondary aim. This is followed by possible alternate hypotheses to explain the observed results. We follow this up with potential causal hypotheses. We then discuss all our findings in terms of support,

care, and rights of transgender and gender diverse individuals, followed by the limitations and conclusions.

****REVIEWERS' COMMENTS:**

Reviewer #3 (Remarks to the Author):

I have studied the authors responses to the concerns I have raised and the changes in the manuscript and I am now content with the recommendation to accept for publication. The authors have given a great deal of thought to the concerns I have raised and have conducted further analyses. They acknowledge that their study is limited by having been primarily undertaken and designed to look at the association between gender diversity and autism and not to look at other neuropsychiatric conditions leaving questions unanswered and have acknowledged the need for further research.

Reviewer #4 (Remarks to the Author):

The authors have been very responsive to previous comments from reviewers. I have no additional comments and believe the revised paper is suitable for publication.

Sincerely,
Lisa D Wiggins